# Genetic diversity of CHC22 clathrin impacts its function in glucose metabolism

Matteo Fumagalli[1,2,3,4†], Stephane M Camus[2†], Yoan Diekmann[3,4†], Alice Burke[2], Marine D Camus[2], Paul J Norman[5,6], Agnel Joseph[7], Laurent Abi-Rached[8], Andrea Benazzo[9], Rita Rasteiro[10], Iain Mathieson[11], Maya Topf[7], Peter Parham[12,13], Mark G Thomas[3,4], Frances M Brodsky[2,7]*

[1]Department of Life Sciences, Imperial College London, Ascot, United Kingdom; [2]Research Department of Structural and Molecular Biology, Division of Biosciences, University College London, London, United Kingdom; [3]Research Department of Genetics, Evolution and Environment, Division of Biosciences, University College London, London, United Kingdom; [4]UCL Genetics Institute, University College London, London, United Kingdom; [5]Division of Bioinformatics and Personalized Medicine, University of Colorado, Aurora, United States; [6]Department of Microbiology and Immunology, University of Colorado, Aurora, United States; [7]Institute of Structural and Molecular Biology, Birkbeck College and University College London, London, United Kingdom; [8]Aix-Marseille Univ, IRD, MEPHI, IHU Méditerranée Infection, CNRS, Marseille, France; [9]Department of Life Sciences and Biotechnology, University of Ferrara, Ferrara, Italy; [10]School of Biological Sciences, University of Bristol, Bristol, United Kingdom; [11]Department of Genetics, Perelman School of Medicine, University of Pennsylvania, Philadelphia, United States; [12]Department of Structural Biology, Stanford University, Stanford, CA, United States; [13]Department of Microbiology and Immunology, Stanford University, Stanford, CA, United States

*For correspondence:
f.brodsky@ucl.ac.uk

†These authors contributed equally to this work

Competing interests: The authors declare that no competing interests exist.

**Abstract** CHC22 clathrin plays a key role in intracellular membrane traffic of the insulin-responsive glucose transporter GLUT4 in humans. We performed population genetic and phylogenetic analyses of the CHC22-encoding *CLTCL1* gene, revealing independent gene loss in at least two vertebrate lineages, after arising from gene duplication. All vertebrates retained the paralogous *CLTC* gene encoding CHC17 clathrin, which mediates endocytosis. For vertebrates retaining *CLTCL1*, strong evidence for purifying selection supports CHC22 functionality. All human populations maintained two high frequency *CLTCL1* allelic variants, encoding either methionine or valine at position 1316. Functional studies indicated that CHC22-V1316, which is more frequent in farming populations than in hunter-gatherers, has different cellular dynamics than M1316-CHC22 and is less effective at controlling GLUT4 membrane traffic, altering its insulin-regulated response. These analyses suggest that ancestral human dietary change influenced selection of allotypes that affect CHC22's role in metabolism and have potential to differentially influence the human insulin response.
DOI: https://doi.org/10.7554/eLife.41517.001

## Introduction

Clathrin-coated vesicles (CCVs) are key players in eukaryotic intracellular membrane traffic (**Brodsky, 2012**). Their characteristic lattice-like coat is self-assembled from cytoplasmic clathrin proteins,

**eLife digest** When we eat carbohydrates, they are digested into sugars that circulate in the blood to provide energy for the brain and other parts of the body. But too much blood sugar can be poisonous. The body regulates blood sugar balance using the hormone insulin, which triggers the removal of sugar from the blood into muscle and fat cells. This removal process involves a pore in membranes at the surface of muscle and fat tissue, called a glucose transporter, through which the sugar molecules can pass. During fasting, the glucose transporter remains inside muscle and fat. But after a meal, insulin acts to release the transporter from its storage area to the surface of the tissue. How efficiently this process happens reflects how efficiently sugar can be removed from the blood. When this pathway breaks down, it can lead to diabetes.

In humans, a protein called CHC22 is needed to deliver the glucose transporter to its storage area. In mice, CHC22 is absent. The question arises: do different animals' eating habits influence CHC22's role in controlling blood sugar? The evolutionary history of CHC22 in a number of different animals could reveal what is special about glucose transport after a meal in humans, and how it might fail in diabetes.

By analyzing the genomes of several different species, Fumagalli et al. found that the gene encoding CHC22 first evolved around the time animals began developing a backbone and complex nervous systems. Afterwards, it was lost by some animals – including mice, sheep and pigs. Fumagalli et al. also discovered that CHC22 varies between individual people. A new form of CHC22, which first appeared in ancient humans, is less effective at holding the glucose transporter inside muscle and fat – leading to a tendency to reduce blood sugar levels. This new form became more common in humans over a period witnessing the introduction of cooking, and later farming; both of these technologies are associated with increased sugar in the diet. But not everyone has this new variant of the gene – both the old and newer variants are present in people today.

The history of CHC22 suggests that it was useful for early humans to hold the glucose transporter inside muscle and fat, keeping blood sugar levels high, which contributed to the development of a large brain. But as humans became exposed to higher dietary levels of sugar the newer form of CHC22 allowed blood sugar to be lowered more readily. People with different forms of CHC22 are likely to differ in their ability to control blood sugar after a meal. In some cases, this could lead to heightened blood sugar levels, which in turn can lead to diabetes.

DOI: https://doi.org/10.7554/eLife.41517.002

captures membrane-embedded protein cargo and deforms the membrane into a vesicle. This process enables CCVs to mediate protein transport to and from the plasma membrane and between organelles. The triskelion-shaped clathrin molecule is formed from three identical clathrin heavy chain (CHC) subunits. Humans have two genes (*CLTC* and *CLTCL1*) that respectively encode CHC17 and CHC22 clathrins (*Wakeham et al., 2005*). CHC17 clathrin, which has three bound clathrin light chain (CLC) subunits, is expressed uniformly in all tissues and forms CCVs that control receptor-mediated endocytosis, as well as lysosome biogenesis and maturation of regulated secretory granules. These pathways are conventionally associated with clathrin function and are mediated by clathrin in all eukaryotic cells (*Brodsky, 2012*). In humans CHC22 clathrin is most highly expressed in muscle and adipose tissue and forms separate CCVs that are not involved in endocytosis (*Dannhauser et al., 2017*). In these tissues, CHC22 CCVs regulate targeting of the glucose transporter 4 (GLUT4) to an intracellular compartment where it is sequestered until released to the cell surface in response to insulin (*Vassilopoulos et al., 2009*). This insulin-responsive GLUT4 pathway is the dominant mechanism in humans for clearing blood glucose into muscle and fat tissues after a meal (*Shepherd and Kahn, 1999*). In addition to its distinct tissue expression pattern and biological function, CHC22 does not bind the CLC subunits that associate with CHC17 clathrin, even though the CHC protein sequences are 85% identical (*Dannhauser et al., 2017*; *Liu et al., 2001*). This remarkable biochemical and functional divergence evolved since the gene duplication event that gave rise to the two different clathrins during the emergence of chordates (*Wakeham et al., 2005*). Notably, however, the *CLTCL1* gene encoding CHC22 evolved into a pseudogene in the *Mus* genus, although mice maintain an insulin-responsive GLUT4 pathway for clearing blood glucose. This

observation suggests that, despite the importance of the *CLTCL1* gene product, backup pathways have evolved to compensate for loss of the CHC22 protein. To understand the evolution of the specialized function of CHC22, and the potential selective processes involved, we here explore the phylogenetic history of the *CLTCL1* gene in vertebrates and its population genetics in humans, non-human primates and bears.

Ecological shifts create selective forces that filter variation in cellular genes. These include changes in nutritional conditions (*Babbitt et al., 2011*), as well as encounters with pathogens (*Fumagalli et al., 2011*); both documented as selective forces that affect membrane traffic genes (*Elde and Malik, 2009*; *Liu et al., 2014*). Recent studies of the evolution of genes involved in membrane traffic have focused on an overview of all eukaryotes with the goals of establishing the origins of membrane-traffic regulating proteins in the last common eukaryotic ancestor and defining the species distribution of various families of traffic-regulating proteins (*Rout and Field, 2017*; *Dacks and Robinson, 2017*). These studies have identified common features of proteins that regulate membrane traffic (*Rout and Field, 2017*) and revealed that extensive gene duplication has allowed lineage-specific diversification of coat proteins and other membrane traffic regulators, such as the Rab GTPases (*Diekmann et al., 2011*; *Guerrier et al., 2017*). Our earlier study of available annotated genomes in 2005 suggested that the gene duplication giving rise to the two CHC-encoding genes occurred as a result of one of the whole genome duplications contributing to vertebrate evolution (*Wakeham et al., 2005*). Here we focus on the more recent evolutionary history of these genes, as well as analyze the increased number of fully annotated vertebrate genomes. We establish that the loss of *CLTCL1* in the *Mus* genus is not unique in vertebrates, identifying at least one additional independent gene loss event in the clade of Cetartiodactyla affecting pigs, cows, sheep, porpoise, and possibly additional related species. Nonetheless, there is strong evidence for CHC22 sequence conservation amongst those species that retain *CLTCL1* (*Wakeham et al., 2005*). This evolutionarily recent gene loss in some lineages and retention of the functional form in others suggested that *CLTCL1* may still be under purifying selection, so we examined *CLTCL1* variation between individuals within vertebrate populations. Comparing populations, we found *CLTCL1* to be considerably more polymorphic than *CLTC*, which encodes the clathrin found in all eukaryotes, with evidence for strong ancient purifying selection for CHC17 clathrin function and relaxed purifying selection on CHC22 function. Additionally, we identified two common allotypes of human CHC22, which have different functional properties. The derived allele arose in ancient humans and is more frequent in farming populations when compared to hunter-gatherers. We previously observed that CHC22 accumulates at sites of GLUT4 retention in the muscle of insulin-resistant patients with type two diabetes (*Vassilopoulos et al., 2009*) in addition to its active role in membrane traffic of GLUT4. Thus, CHC22 variation has potential to differentially affect membrane traffic pathways involved in insulin resistance, as well as alter normal glucose metabolism within human and other vertebrate populations. The analyses reported here lead us to propose that variation in the CHC22 clathrin coat may be a response to changing nutritional pressures both between and within vertebrate species.

## Results

### Phylogenetic analyses reveal selective loss or retention of a functional *CLTCL1* gene in vertebrates

Identification of CHC-encoding genes in 62 vertebrate and non-vertebrate species ((*Figure 1—figure supplement 1*, *Figure 1—figure supplement 2*) indicates a dynamic history of gene duplications and losses (*Figure 1*). The *CLTCL1* gene was detected only in jawed vertebrates (bony vertebrates and cartilaginous fish), while the two jawless vertebrate genomes available – lamprey (*Petromyzon marinus*) and hagfish (*Eptatretus burgeri*) – have only one CHC-encoding gene. This distribution refines the timing of the CHC-encoding gene duplication to the period after the Agnatha split off the vertebrate lineage, estimated at 493.8MYA (95% HPD: 459.3, 533.8), and before the evolution of jawed vertebrates 450.8MYA (95% HPD: 432.1, 468.1) (*Hedges et al., 2015*; *dos Reis et al., 2015*). Of the ten species of bony fish that split off the spotted gar (*Lepisosteus oculatus*) lineage (*Amores et al., 2011*), whose genomes are generally tetraploid, all had two versions of *CLTC* and at least one *CLTCL1* gene, except for cave fish (*Astyanax mexicanus*) apparently lacking *CLTCL1*. Eight additional species of vertebrates with high genome coverage and reliable annotation had the

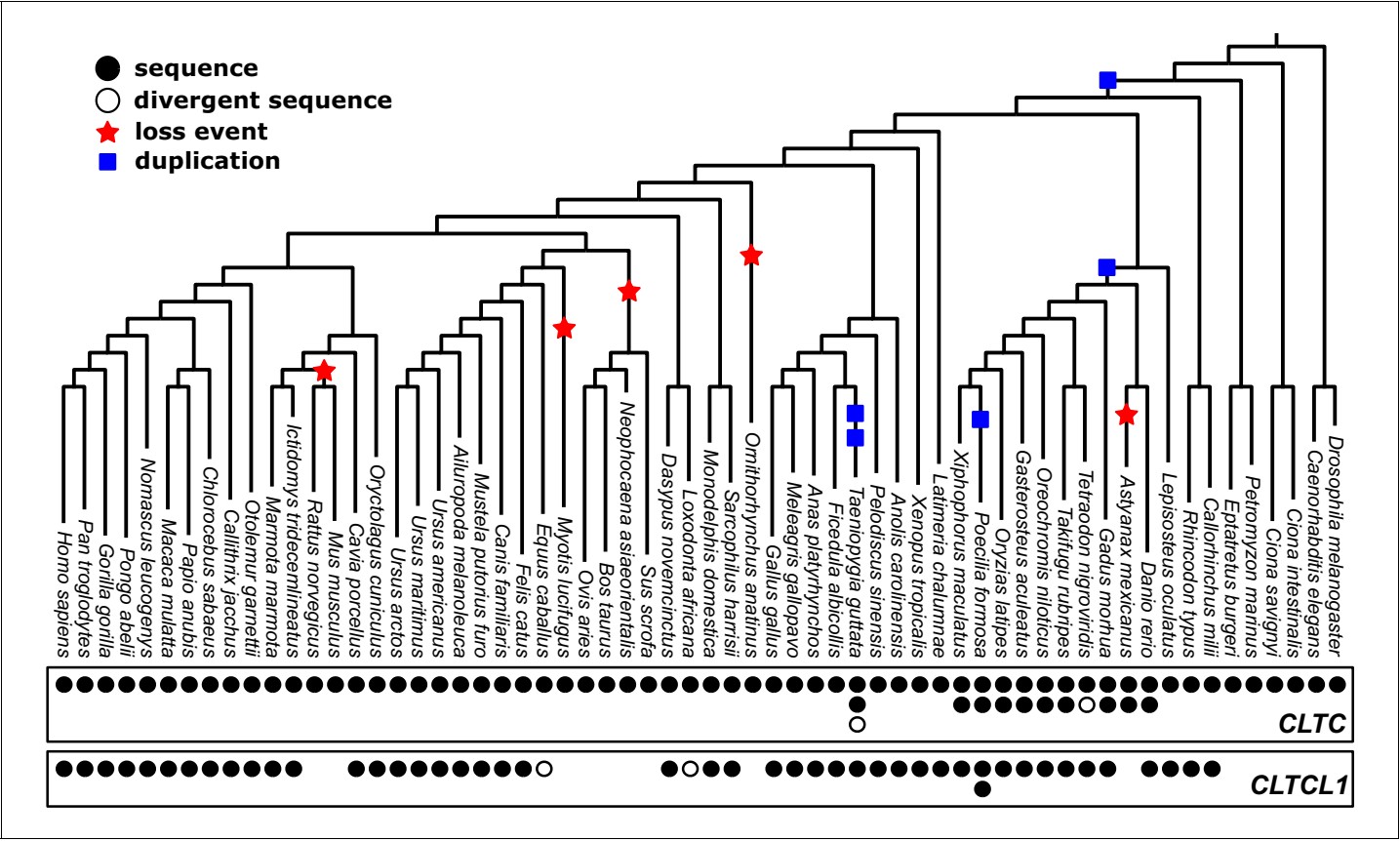

**Figure 1.** Phylogenetic analysis of *CLTC/CLTCL1* reveals independent loss of the gene encoding CHC22 clathrin from vertebrate lineages and complete conservation of the gene encoding CHC17 clathrin. Phylogenetic profiles *of CLTC/CLTCL1* are shown, with gene presence in the corresponding genome indicated by a filled black circle. All sequences used have less than 5% unspecified residues ('X's in the relevant database). Divergent gene sequences with low sequence similarity but that still fall within the *CLTC* clade are shown as empty circles (see Materials and methods for similarity threshold and sequence IDs). Based on the profile and species tree the most parsimonious phylogenetic tree for loss and duplication events is inferred and shown as red stars and blue squares, respectively.

DOI: https://doi.org/10.7554/eLife.41517.003

The following figure supplements are available for figure 1:

**Figure supplement 1.** Unreconciled phylogenetic tree for *CLTC* and *CLTCL1* across all investigated species.
DOI: https://doi.org/10.7554/eLife.41517.004

**Figure supplement 2.** Reconciled phylogenetic tree (therefore missing support values) for *CLTC* and *CLTCL1* across all investigated species.
DOI: https://doi.org/10.7554/eLife.41517.005

*CLTC* gene but no identifiable *CLTCL1* gene. *CLTCL1* genes are present in the Caviomorpha and Sciuridae rodent suborders, and lost in the Muroidea suborder from the entire *Mus* genus (**Wakeham et al., 2005**) and from rat (*Rattus norvegicus*). The Cetartiodactyla clade also appears to have lost *CLTCL1*, as *CLTCL1* is absent from the four representative genomes in our dataset (pig (*Sus scrofa*), sheep (*Ovis aries*), cow (*Bos taurus*), Yangtze finless porpoise (*Neophocaena asiaeorientalis*)). This suggests a loss event before the Cetartiodactyla lineage split, independent of the loss event preceding split of the Muroidea lineage. The absence of *CLTCL1* in rat clarifies why CHC22 could not be biochemically identified in rat and indicates that antibodies against CHC22 that react with rat cells must cross-react with other proteins (**Towler et al., 2004**). *CLTCL1* was also not detected in the genomes of the little brown bat (*Myotis lucifugus*) and the duck-billed platypus (*Ornithorhynchus anatinus*). Assuming the genome annotations for the species analyzed are reliable, these data indicate that there have been at least five independent losses of *CLTCL1* that are clade- or species-specific. The intermittent loss of *CLTCL1* and the retention of *CLTC* raises the question of

whether their patterns of evolution are typical for genes with related functions that duplicated in the same time frame as *CLTC/CLTCL1*.

*CLTC* and *CLTCL1* are located on paralogous regions of human chromosomes 17 and 22, respectively. For these two genes, the evolutionary rates (rate of non-synonymous substitutions to rate of synonymous substitutions; dN/dS) across vertebrates at each position were determined and plotted

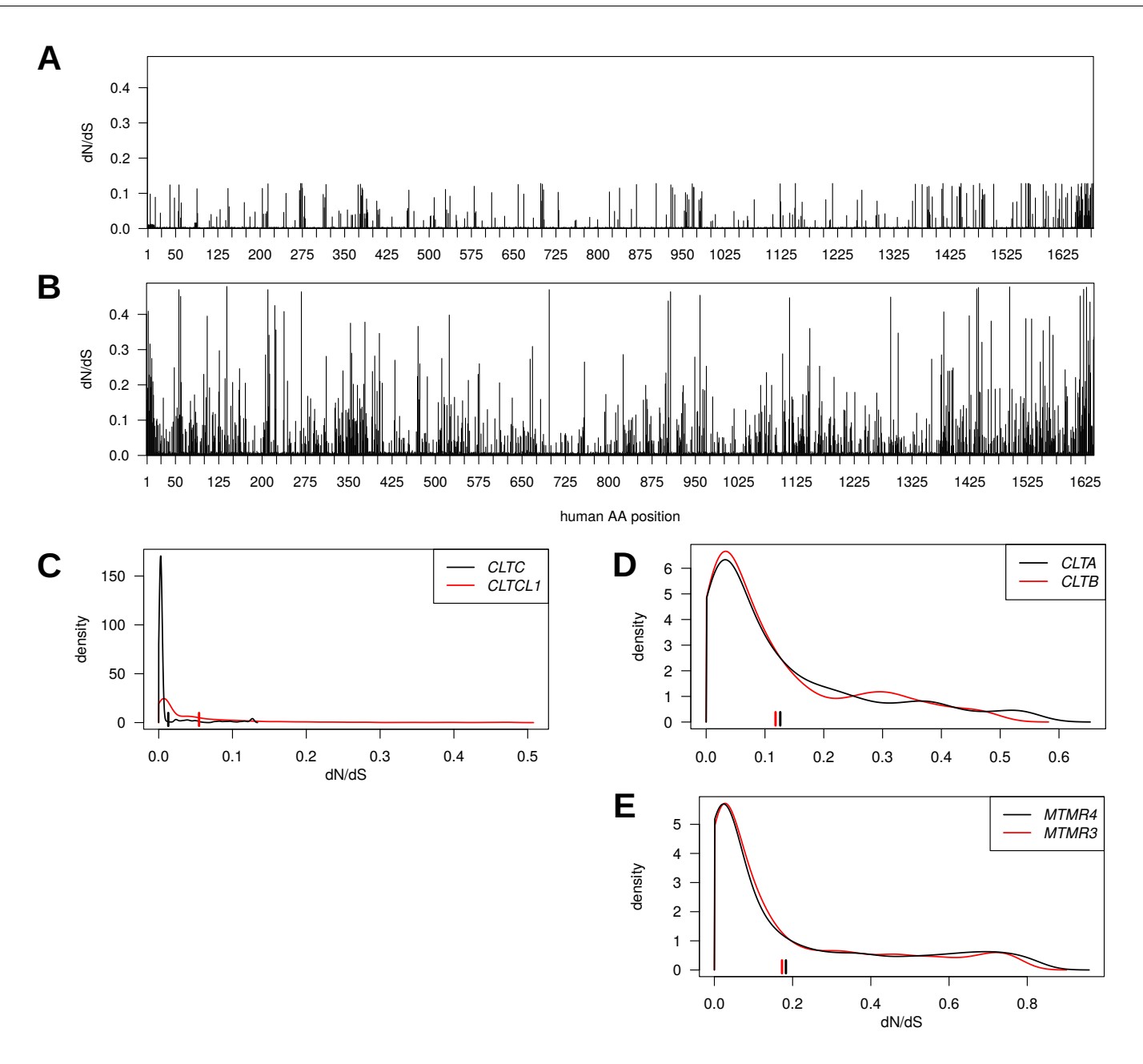

**Figure 2.** Genes encoding clathrin heavy chains show evidence for purifying selection with *CLTCL1* (CHC22-encoding) being more variable than *CLTC* (CHC17-encoding) over evolutionary time. Evolutionary rates expressed as dN/dS ratios are shown for each position in *CLTC* (A) and *CLTCL1* (B). Rates are averages over an entire phylogenetic tree and therefore not specific to the human proteins. However, to assist interpretation, only rates for residues present in the human proteins are shown. Kernel density estimates of the distributions of dN/dS ratios per paralogous pair of proteins (C–E). *CLTA* and *CLTB* encode clathrin light chains A and B, respectively. *MMTR3* and *MMTR4* encode myotubularin lipid phosphatases. Mean dN/dS ratios averaged over all sites are shown as hatched marks.

DOI: https://doi.org/10.7554/eLife.41517.006

along the length of the protein sequences (*Figure 2A–B*). Several adjacent paralogs have been maintained in these chromosomal regions, some of which are involved in membrane traffic, including the gene pair of *MTMR4* and *MTMR3*, encoding myotubularin lipid phosphatases. Also, CLC subunits of CHC17 clathrin are encoded by paralogous genes on different chromosomes (*CLTA* and *CLTB*) that arose from a local gene duplication, mapped to the same time frame as the CHC-encoding duplication (*Wakeham et al., 2005*). Comparison of the distribution of dN/dS ratios for the three pairs revealed stronger purifying selection on the *CLTC/CLTCL1* genes than on *MTMR4/MTMR3* and *CLTA/CLTB* (*Figure 2C–E*), suggesting the CHC-encoding clade is more evolutionarily constrained. This observation is consistent with our previous identification of conserved signature residues in *CLTCL1* using DIVERGE analysis (*Wakeham et al., 2005*) and indicates conserved functions for both the *CLTC* and *CLTCL1* gene products. Furthermore, there is a striking difference in the distribution and average of evolutionary rates, as measured by dN/dS, between *CLTC* and *CLTCL1* (Kolmogorov-Smirnov test p-value<2.2e-16), with *CLTC* being significantly more constrained by purifying selection than *CLTCL1*. In contrast, there is minimal difference in the distribution and average of evolutionary rates between the two paralog pairs *MTMR4/MTMR3* and *CLTA/CLTB* (Kolmogorov-Smirnov test yields *p*-values 0.003643 and 0.9959, respectively).

## Human population genetic analyses indicate purifying selection with ongoing diversification for *CLTCL1*

To follow up the indication that *CLTC* and *CLTCL1* are subject to different degrees of purifying selection, we investigated their variation in human populations. We analyzed 2504 genomes from the 1000 Genomes Project database, phase 3 (1000 *Auton et al., 2015*) and identified alleles resulting from non-synonymous substitutions for *CLTC* and *CLTCL1*. This dataset included individuals from each of five human meta-populations: European (EUR, 503), East Asian (EAS, 504), Admixed American (AMR, 347), South Asian (SAS, 489) and African (AFR, 661). Individual populations with their abbreviations are listed in *Supplementary file 1a*. The reference sequences for chimpanzee (*Pan troglodytes*) and pseudo-references for two archaic humans, Altai Neanderthal and Denisovan, were also included to relate allelic variation to the ancestral state. A median-joining network for all the inferred *CLTC* human alleles showed a very common allele (sample frequency 0.997) with only five low-frequency variants generating a total of six alleles (*Figure 3A*). Each allele encodes a variant of a CHC (allotype), which includes one or more single nucleotide polymorphisms (SNPs).

In contrast to *CLTC*, we identified 46 non-synonymous SNPs in *CLTCL1*, present in 52 distinct haplotypes (referred to here as alleles, following the definition given above, *Supplementary file 2b-c*). A median-joining network for the most common *CLTCL1* alleles showed that they are widely distributed within the human meta-populations (*Figure 3B*). Each meta-population tends to have private, less frequent alleles. Nevertheless, all the meta-populations comprised two main allelic clades, together constituting a sample frequency of 77%. These two main alleles differ by a single methionine to valine substitution at position 1316 (M1316V) in the protein sequence (SNP ID rs1061325 with genomic location chr22:19184095 on hg19 assembly). The valine at position 1316 is predicted to have a functional effect on the protein since it was categorized as 'probably damaging' with a probability of 0.975 by PolyPhen (*Adzhubei et al., 2010*) and as 'damaging' by SIFT (*Kumar et al., 2009*). Sequences in chimpanzee and archaic humans have the M1316 allotype, suggesting that M1316 is likely to represent the ancestral state. To further investigate this, we inspected raw sequencing data from both Altai Neanderthal and Denisovan (*Supplementary file 1d*). We inferred the most likely genotype to be homozygous for the M1316 amino acid (minimum sequencing depths equal to 40 and 28, respectively). We then extracted sequencing data for an additional 13 archaic and ancient humans (*Supplementary file 1d*). We found that the V1316 amino acid is present in Pleistocene hunter-gatherers and Neolithic farmers but not in other Neanderthals or Holocene hunter-gatherers in this limited data set. The equivalent residue in human CHC17 (encoded by *CLTC*) is also methionine, suggesting that methionine at this position likely pre-dated the initial duplication generating *CLTCL1*. For the non-human species analyzed (*Figure 1*), all CHC-encoding genes present would produce clathrins with M1316, further indicating its ancient and conserved role in CHC structure or function.

To quantify the levels of nucleotide and allelic diversity for non-synonymous sites within human populations, several summary statistics of diversity were calculated. For populations within each meta-population, we separately calculated Watterson's and Nei's estimators of genetic diversity (TW

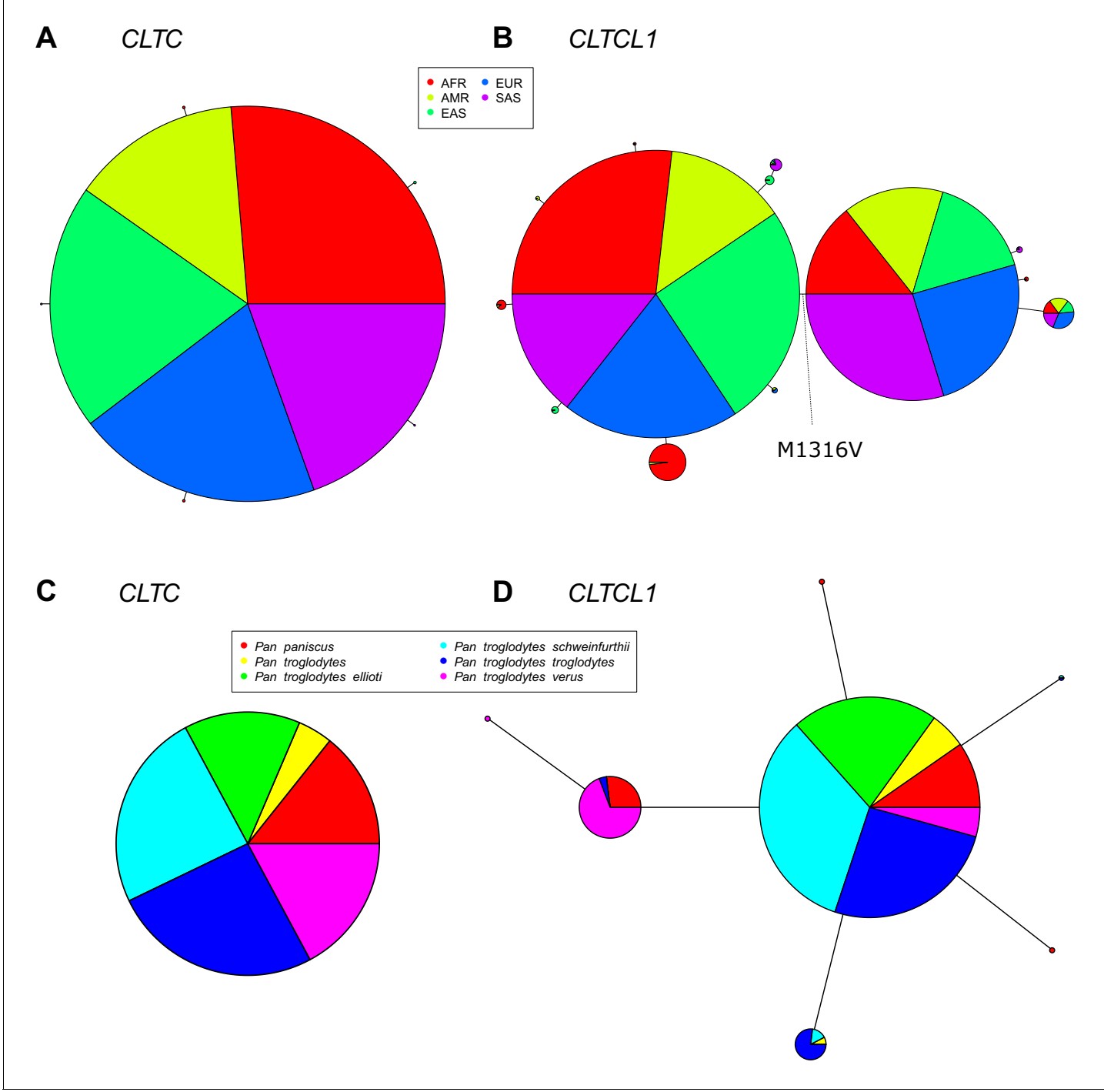

**Figure 3.** The *CLTCL1* gene encoding human CHC22 has two major variants, and is highly polymorphic relative to the human *CLTC* gene encoding CHC17, with a similar pattern in chimpanzees. Median joining network of human alleles for *CLTC* (A) and *CLTCL1* (B) are shown. Each circle represents a unique allele whose global frequency is proportional to its circle's size and the line length between circles is proportional to the number of non-synonymous changes between alleles. For *CLTC*, the least common alleles have a frequency ranging from 0.04% and 0.06% and the circles representing them were magnified by a factor of 10. For *CLTCL1*, only alleles with a frequency greater than 20% were plotted. The two major alleles show a combined frequency of 77% while the other alleles depicted in the figure have a frequency ranging from 0.44% to 5.67%. Segregation of the M1316V variation is depicted with a hashed line, with alleles carrying the M variant on the left-hand side, and alleles carrying the V variant on the right-hand side. The meta-populations in which the allele is found are indicated in color representing their percentage of the total frequency of the allele in humans. Meta-populations analyzed are African (AFR), American (AMR), East Asian (EAS), European (EUR), South Asian (SAS). (C–D) Median joining network of *CLTC* (C) and *CLTCL1* (D) alleles for chimpanzees (*Pan troglodytes*, four identified subspecies and one unidentified) and bonobos (*Pan*

*Figure 3 continued on next page*

*Figure 3 continued*

*paniscus*). The species and subspecies in which each variant is found are indicated in color representing their percentage of the of the total frequency of the variant in chimpanzees and bonobos.

DOI: https://doi.org/10.7554/eLife.41517.007

The following figure supplement is available for figure 3:

**Figure supplement 1.** Phylogenetic trees of amino acid sequences for *CLTC* and *CLTCL1* in the bear samples analyzed.

DOI: https://doi.org/10.7554/eLife.41517.008

and PI, respectively), Tajima's D (TD), Fu and Li's D* (FLDs) and F* (FLFs), the sum of squared allele frequencies including the most common allele (H1) and excluding it (H2), and the normalized ratio between H2 and H1 (H2H1) (*Supplementary file 2a*). To assess whether observed summary statistics are expected or not under neutral evolution in each population, we calculated the empirical null distribution from a set of 500 control genes with the same coding length as *CLTCL1* (*Supplementary file 2b*). High or low percentile rank values for *CLTCL1* in the empirical distribution indicate that the summary statistic for *CLTCL1* is unlikely to occur by mutation and neutral genetic drift alone. Summary statistics and populations were then clustered according to their empirical ranks and plotted on a heat map (*Figure 4*).

All populations tend to display high genetic diversity for *CLTCL1*, as summarized by PI and TW, and an unusually high frequency for the second most common allele, as summarized by H2 and H2H1. Such configuration is likely to occur under balancing selection (*Charlesworth, 2006*) or because of a soft sweep (*Messer and Petrov, 2013*). That *CLTCL1* was low ranking in all populations for H1, a statistic representing the frequency of the most common allele, also supported diversifying selection rather than hard sweeps. On the other hand, all populations display negative TD values with many populations exhibiting negative FLDs and FLFs values. These values are consistent with low diversity within common alleles and an excess of low-frequency variants. Finally, we calculated a measure of genetic differentiation (fixation index $F_{ST}$) between pairs of canonical reference populations, namely Yoruba from Nigeria (YRI), North Americans with European ancestry (CEU), and Han Chinese from Beijing (CHB). We did not find any evidence that $F_{ST}$ values for *CLTCL1* (YRI-CEU 0.15, YRI-CHB 0.077, CEU-CHB 0.065) are outliers in the empirical distribution of control genes.

Such inconsistent patterns could be partly explained by the fact that we considered only non-synonymous changes, and the limited number of SNPs considered per gene may create a larger variance in the empirical distributions, especially for allele-based statistics. We therefore further examined the high frequency of the second most common allele by investigating whole genomic variation, including silent SNPs. We observed a local increase of H2 statistics in *CLTCL1* for European populations, which already shows a large value based on non-synonymous changes (*Figure 4—figure supplement 1*). This analysis also indicates that any selection signatures are restricted to a local genomic region encompassing *CLTCL1*.

Another reason for the summary statistics not being strong outliers in the empirical distribution is the high recombination rate (sex-average rate of 2.5 cM/Mb) inferred for the genomic region encompassing *CLTCL1* (*Kong et al., 2002*). We therefore performed coalescent simulations under neutrality of a putative 100kbp genomic region surrounding the SNP encoding the M1316V variation, taking into account the local recombination rate and a previously proposed demographic model for Africans (YRI), Europeans (CEU) and East Asians (CHB) (*Gutenkunst et al., 2009*) with a mutation rate of $1.5 \times 10^{-8}$ per base pair per generation. The observed values for TW and PI were significantly greater than expected under neutral evolution for all populations (p-values<0.001), while TD was greater than expected for CHB only, although with a marginally non-significant statistical support (*p*-value 0.056). All these results are suggestive of a genetic diversity higher than expected under neutrality for a region encompassing M1316V, although possible complex evolutionary scenarios may limit the power of summary statistics to detect such selective events.

One plausible explanation of the high genetic diversity and frequency of the two major alleles of *CLTCL1* that occur in all modern human populations (*Figure 3B*, *Supplementary file 1b*) is balancing selection (*Charlesworth, 2006*). Such a distribution of allele frequency was confirmed using a different data set of more than 50 sampled human populations (*Figure 4—figure supplement 2*). In several populations, we also observed an apparent excess of heterozygosity at SNP rs1061325

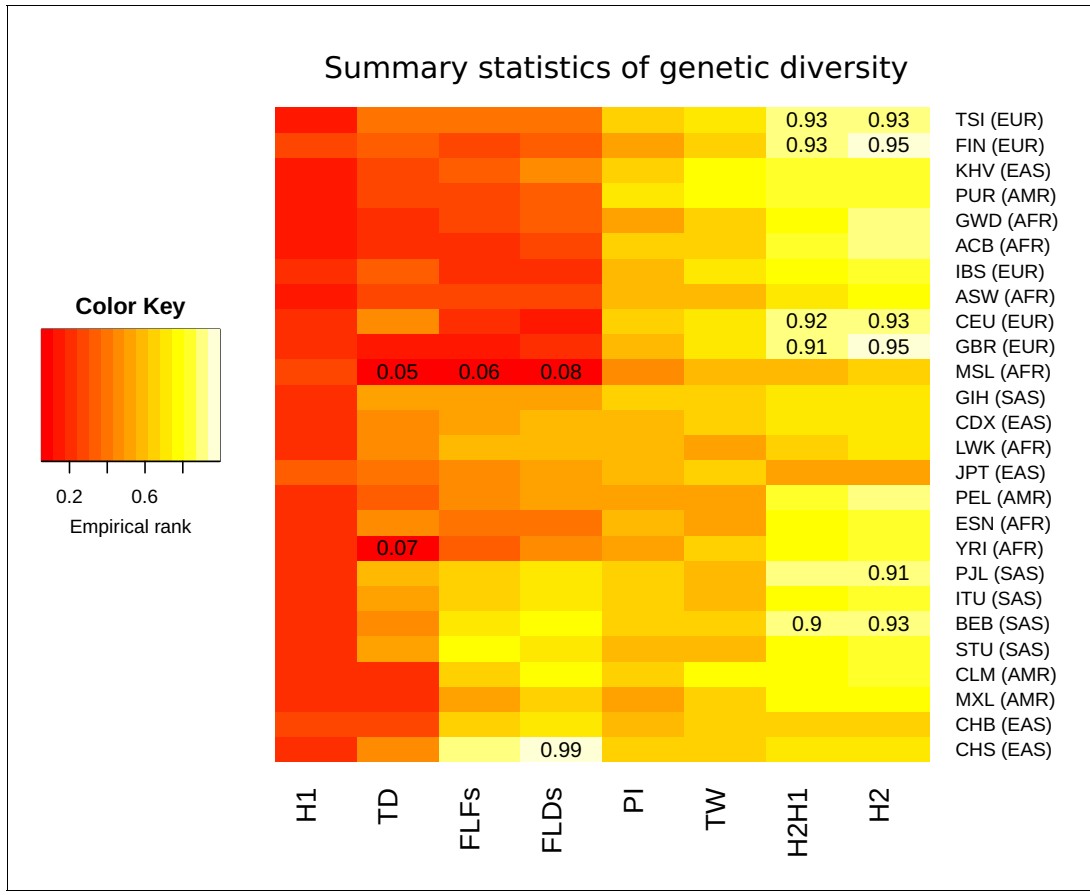

**Figure 4.** Summary statistics for genetic diversity of *CLTCL1* indicate selection over neutral variation. For each human population (on the rows) we calculated several summary statistics to analyze diversity (on the columns, defined in Materials and methods) and reported their percentile rank against their corresponding empirical distribution based on 500 control genes. The resulting matrix was then sorted on both axes as a dendrogram (not reported) based on the pairwise distances between each pair of populations. The populations analyzed, with their abbreviations, are listed in *Supplementary file 1a*, and the inclusive meta-population is indicated in parentheses, defined as in the legend to *Figure 3*. As depicted in the color legend, red and yellow denote low and high percentile ranks, respectively. Percentiles lower than 0.10 or greater than 0.90 are given in the corresponding cell.

DOI: https://doi.org/10.7554/eLife.41517.009

The following figure supplements are available for figure 4:

**Figure supplement 1.** Variation of H2 statistics along a genomic region surrounding *CLTCL1* in four European populations, with abbreviations as defined in *Supplementary file 1a*.
DOI: https://doi.org/10.7554/eLife.41517.010

**Figure supplement 2.** Geographical distribution of M1316- and V1316-encoding alleles across human populations in the HGDP-CEPH panel data set.
DOI: https://doi.org/10.7554/eLife.41517.011

**Figure supplement 3.** Worldwide distribution of heterozygosity of M1316- and V1316-encoding alleles.
DOI: https://doi.org/10.7554/eLife.41517.012

(*Supplementary file 2c*), compatible with heterozygote advantage (overdominance) for the two encoded allotypes differing at residue 1316. Specifically, all European populations show a ratio of observed versus expected (assuming Hardy-Weinberg equilibrium) heterozygosity greater than 1, with the highest value of 1.24 (chi-squared test nominal *p*-value 0.047) for Iberic Spanish (IBS) (*Figure 4—figure supplement 3*). Selective pressures that might be acting on *CLTCL1*, irrespective of population distribution, could be changes in human diet, a number of which have been inferred over the last 2.6 million years (*Hardy et al., 2015*). Perhaps the best known of these dietary transitions are the introduction of cooking ~450 KYA, the development of farming ~12,500 YA, and more recently industrialized food processing, which gradually and then dramatically increased

carbohydrate availability and consumption by humans. As CHC22 clathrin, the gene product of *CLTCL1*, is required for formation of the intracellular pathway critical for an insulin response, its genetic history could potentially be influenced by these changes. To address the hypothesis that nutritional habits conferred selective pressure on *CLTCL1*, we compared the frequency of SNP rs1061325 (M1316V) in farming versus hunter-gatherer population samples from ancient and modern humans. Although the appearance of SNP rs1061325 predates the advent of farming (*Supplementary file 1d*), the observed frequencies of this allele, which encodes the CHC22-V1316 allotype, are consistent with a tendency for it to increase once farming became common practice for a population (*Figure 5*), although the small sample size for modern humans limits the power to reach statistical significance. The highest difference in allele frequency was observed between early farmers and hunter-gatherers from West Eurasia. However, as these two populations are highly diverged, it remains possible that this significant difference in allele frequency is due to genetic drift shaped by population history, rather than natural selection. To test this model, using the same dataset, we extracted 2500 control SNPs with a global minor allele frequency similar to rs1061325 (M1316V) (up to an error of 5%) and minimum global sequencing depth of 100X. We obtained statistical significance (*p*-value 0.036) when testing the difference in derived allele frequency in farmers compared to hunter-gatherers (+26.58%), while we found no statistical support (*p*-value 0.080) when testing the absolute difference in allele frequency between these two populations.

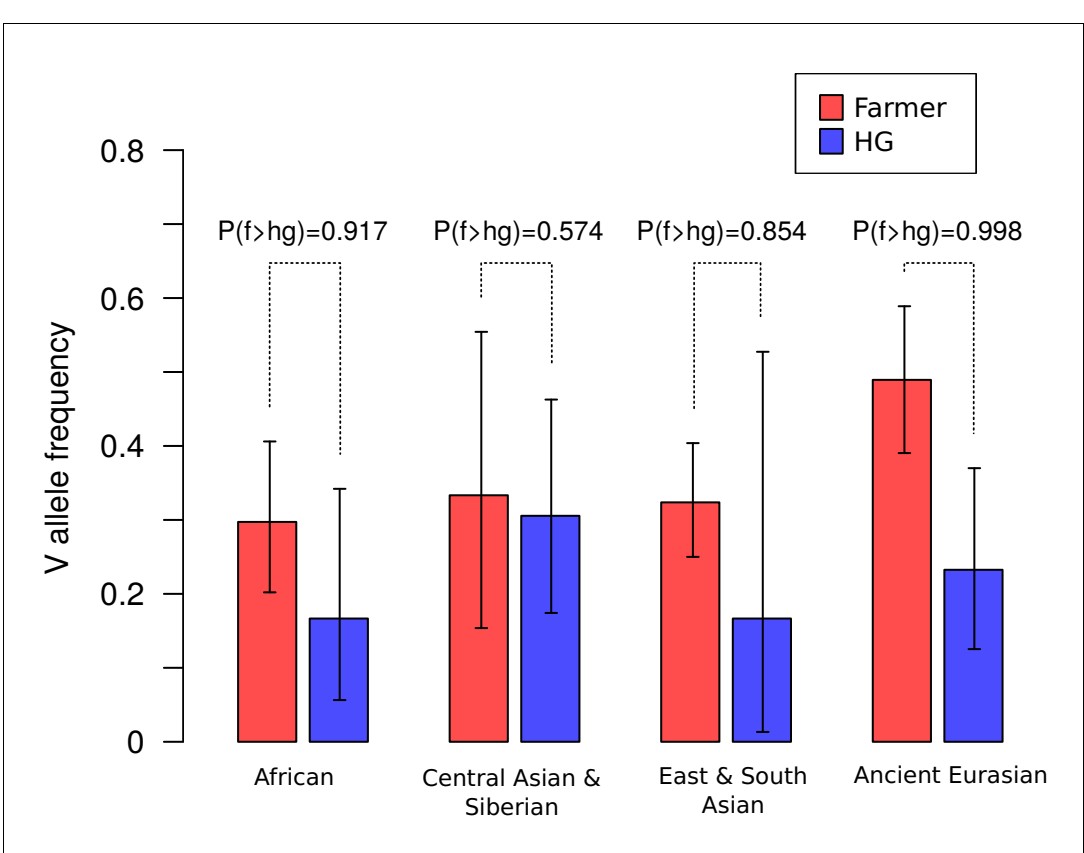

**Figure 5.** Frequencies of the V1316 variant of CHC22 trend higher in populations of farmers compared to hunter-gatherers. Maximum a posteriori estimates and 95% highest posterior density credible intervals of the frequency of V1316 are compared for modern and ancient hunter-gatherer (HG) and farmer populations indigenous to three continents. Probability of the V allele being at a higher frequency in farmers, labeled as P(f > hg), is also reported.
DOI: https://doi.org/10.7554/eLife.41517.013

## Genetic variation in non-human vertebrate species supports functional diversification of *CLTCL1*

We analyzed allelic variation for *CLTC* and *CLTCL1* in the genomes of 79 individuals representing six species of great ape, two species each for chimpanzees, gorillas and orangutans (*Pan troglodytes*, *Pan paniscus*, *Gorilla beringei*, *Gorilla gorilla*, *Pongo abelii*, *Pongo pygmaeus*). After data filtering and haplotype phasing, we found no non-synonymous SNPs for *CLTC* and 64 putative non-synonymous SNPs for *CLTCL1* (*Supplementary file 3a*). In three species of great apes analyzed, one of the non-synonymous changes in *CLTCL1* leads to a premature stop-codon at amino position 41, with an overall frequency of 36%. However, sequences containing the stop-codon exhibited only a marginal increase of nucleotide diversity (+4.7% as measured by Watterson's index; *Watterson, 1975*) compared to the full-length sequences, suggesting that these are relatively new variants. Notably, for all the non-human primates analyzed, *CLTCL1* variants do not encode the V1316 allotype, which appears private to humans. However, in all three types of great ape we found a common but different substitution, threonine (T1316), at the same amino acid position.

To further investigate variation in non-human primates, we increased the sample size per species by analyzing *CLTC* and *CLTCL1* variation in 70 chimpanzee and bonobo genomes, including four subspecies of chimpanzee. While no variation was observed for *CLTC* (*Figure 3C*), a median-joining network for the inferred 8 *CLTCL1* alleles (*Supplementary file 3b*) showed a major allele common to different species and subspecies with less frequent alleles primarily restricted to individual ones (*Figure 3D*). In this chimpanzee data set, we observed considerable diversity, with a potential tendency towards multiple variants. However, amino acid 1316 was not covered in this data set, possibly due to poor data mapping quality associated with the high nucleotide diversity observed. In another data set of 20 individuals (*Teixeira et al., 2015*), we found a frequency of 10% for the T1316 allotype in chimpanzees but not in bonobos.

We further investigated *CLTCL1* variation in polar bears (*Ursus maritimus*) and their closest related species, brown bears (*Ursus arctos*). These two species, which diverged 479–343 KYA (*Liu et al., 2014*), have very different diets (*Liu et al., 2014*; *Bojarska and Selva, 2012*) and are phylogenetically closer to each other than chimpanzees and humans. Polar bears subsist on a high fat, low carbohydrate diet, whereas brown bears consume a more varied diet of carbohydrate, protein and fat. Analysis of 21 bear genomes (seven polar bears and 14 brown bears) (*Benazzo et al., 2017*), revealed three positions (1267, 1389, and 1522) which are fixed in polar bears but are either polymorphic or have a different residue in brown bears (*Supplementary file 3c*). Genetic differentiation between polar and brown bears, as measured by $F_{ST}$, is markedly higher for *CLTCL1* (0.56) than for *CLTC* (0.26) (*Liu et al., 2014*). Furthermore, a phylogenetic tree of both bear species in our sample exhibits more diversification for *CLTCL1* compared to *CLTC* (*Figure 3—figure supplement 1*). This sample of bear populations may support the emergence of multiple *CLTCL1* variants within a species and a potential role for diet-related selection.

## Modeling CHC22 variation based on *CLTCL1* polymorphism suggests an effect on clathrin lattice contacts

One expectation for selection of the human-specific *CLTCL1* allele encoding the CHC22-V1316 allotype is that this amino acid change might confer a functional change in the clathrin lattice. This was predicted by the PolyPhen and SIFT analyses, highlighting the change as potentially structure-altering. As many humans are heterozygous for the M1316 and V1316 allotypes (44% based on all individuals from 1000 Genomes project), there may potentially be special properties for mixed lattices formed from the two protein allotypes. To address the possibility that the M1316V polymorphism affects protein function, we used MODELLER (*Benjamin and Sali, 2014*) to produce a homology model of the two CHC22 allotypes based on the crystal structure of CHC17 clathrin (PDB 1B89) (*Ybe et al., 1999*), taking advantage of the 85% protein sequence identity between human CHC17 and CHC22 (*Figure 6*). Modeling using UCSF Chimera (*Pettersen et al., 2004*) showed that residue 1316 is found at a key interface between triskelion legs in assembled clathrin (*Figure 6A* and top of panel B). If M1316 is substituted by V1316, the smaller side chain creates a void that would be energetically unfavorable (*Figure 6*, bottom of panel B), such that the triskelion leg might twist slightly to close the void. In the clathrin lattice, the legs have a torque that rotates the assembly interface along the protein sequence (*Wilbur et al., 2005*), so a further twist could slightly adjust the

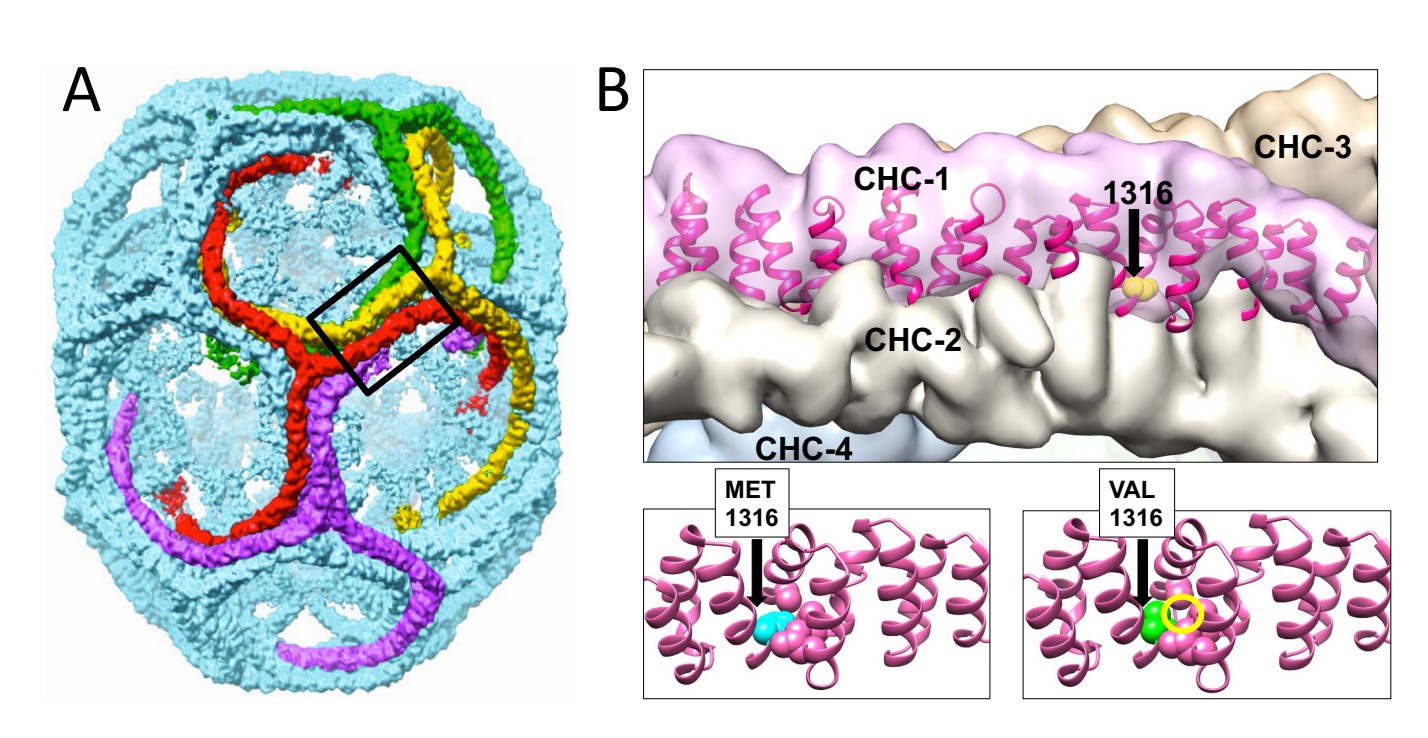

**Figure 6.** Modeling of the structural changes in clathrin caused by the methionine-valine dimorphism at residue 1316 predicts conformational alteration. Model of the CHC17 clathrin lattice (**A**) is reproduced with permission (*Fotin et al., 2004*) with the region comprising residue 1316 boxed. Panel B (top part) is the magnified boxed region in A with a CHC22-M1316 model (residues 1210 to 1516) docked into one of the four clathrin heavy chains (CHC-1) forming the edge of the lattice. The black arrow shows the location of the amino acid residue 1316 with the side chain highlighted in CHC-1. The density of the other three CHCs is indicated. Computational models of human CHC22 (residues 1210 to 1516) with either Met or Val at position 1316 (**B**, lower parts). The yellow circle encloses space opened by reducing the side chain size, which would require a shift in CHC torque to regain structurally favorable side chain contacts.

DOI: https://doi.org/10.7554/eLife.41517.014

interface, altering assembly interactions. Changes in the assembly interface could affect integrity of the lattice and potentially influence kinetics of assembly and disassembly. Mixed lattices of the two CHC22 allotypes would therefore have different properties from CHC22 coats formed in homozygotes for the two major *CLTCL1* alleles. CHC22 is needed for the traffic of GLUT4 to its intracellular storage compartment, where GLUT4 awaits release to the plasma membrane in response to insulin. However, CHC22 also accumulates at the GLUT4 storage compartment (GSC) when it expands due to impaired GLUT4 release in cases of insulin-resistant type two diabetes (T2D) (*Vassilopoulos et al., 2009*). Thus, genetic variation of CHC22 could alter rates of retention and release of GLUT4 in both healthy and disease states.

## CHC22 variants display functional differences

To test whether the evolutionary change from M1316 to V1316 in CHC22 clathrin alters its properties, three aspects of CHC22 biochemistry and function were compared for the two allotypes. HeLa cells were transfected with constructs encoding each CHC22 variant or CHC17, tagged with green fluorescent protein (GFP). Atypically for their epithelial cell origin but not for transformed cells, HeLa cells express CHC22 clathrin (they are homozygous for the M1316 allotype) (*Adey et al., 2013*; *Landry et al., 2013*). We observed that the transfected fluorescently tagged CHC22 allotypes were both concentrated in the perinuclear region of the cell, similar to endogenous CHC22-M1316 detected by antibody, and did not overlap with endogenous CHC17 (*Figure 7A*). Conversely, transfected GFP-CHC17 did not overlap with endogenous CHC22, so expression of the transfected CHCs reflected their natural distribution (*Dannhauser et al., 2017*). Using these constructs, the dynamics

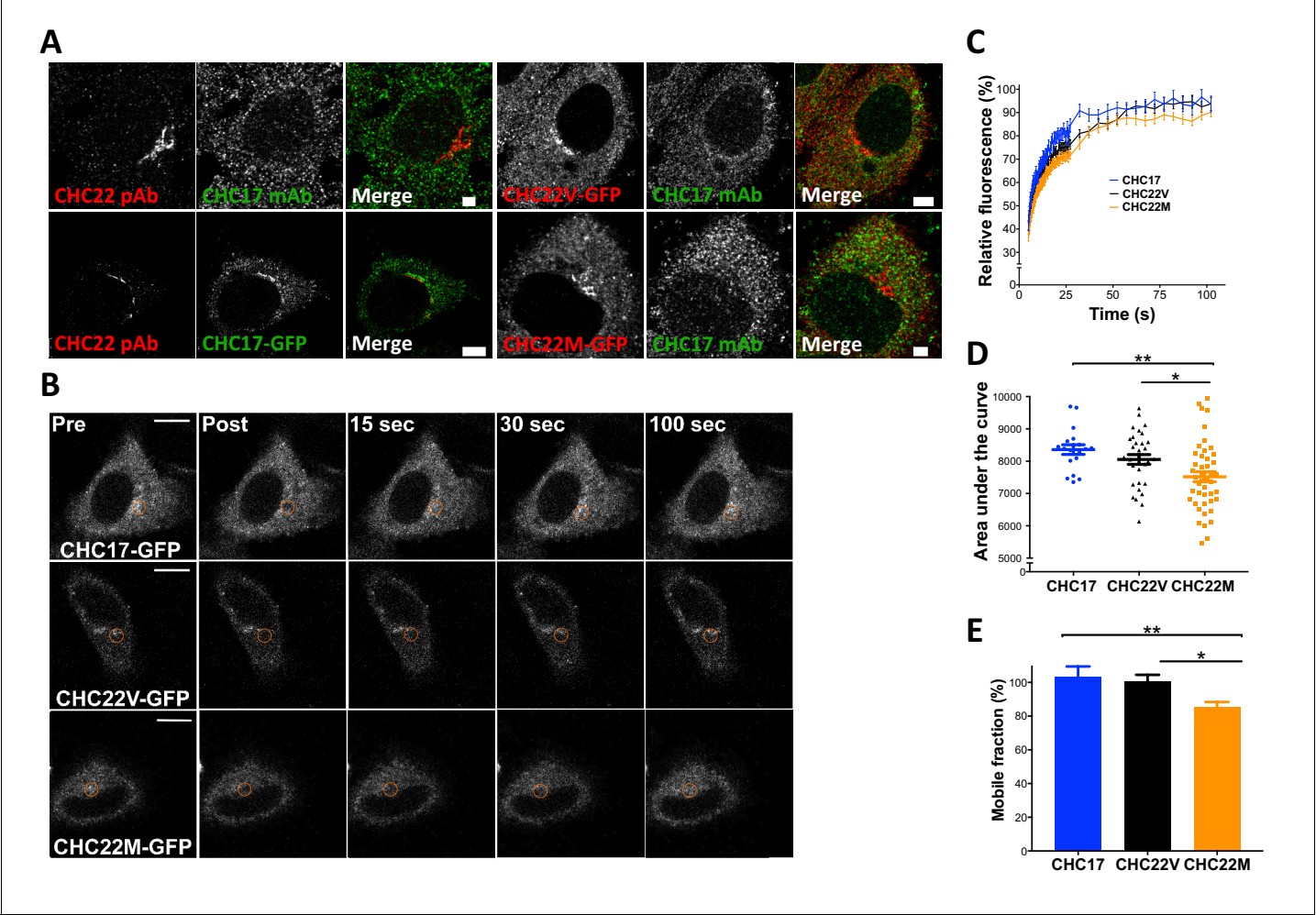

**Figure 7.** The CHC22-M1316 and CHC22-V1316 allotypes have different dynamics of membrane association, as measured by fluorescence recovery after photobleaching (FRAP). HeLa cells were transfected with CHC22-M1316-GFP (CHC22M) or CHC22-V1316-GFP (CHC22V) or CHC17-GFP and the expressed constructs were localized relative to endogenous CHC22 and CHC17, which were also compared to each other (A). Endogenous CHC22, CHC17 and the transfected proteins were visualized by immunofluorescence with anti-CHC22 rabbit polyclonal antibody (pAb, red), anti-CHC17 mouse monoclonal antibody (mAb, green) and anti-GFP chicken polyclonal antibody (green for CHC17-GFP or red for CHC22-GFP), respectively. Bars represent 3 μm (untransfected and CHC22M-GFP) and 5 μm (CHC22V-GFP and CHC17-GFP). Transfectants were photobleached in the circular region indicated (B) and recovery of fluorescence (FRAP) was visualized over time (bars, 10 μm) and quantified within the bleached regions (C). For the data in (C), area under the curves (D) and mobile fractions Mf (E) were calculated (*Lippincott-Schwartz et al., 2001*). We performed a one-way analysis of variance (ANOVA) with Tukey's multiple comparison post-hoc test: * p-value<0.05, ** p-value<0.01.

DOI: https://doi.org/10.7554/eLife.41517.015

The following source data is available for figure 7:

**Source data 1.** FRAP experiment data set 1.
DOI: https://doi.org/10.7554/eLife.41517.016
**Source data 2.** FRAP experiment data set 2.
DOI: https://doi.org/10.7554/eLife.41517.017
**Source data 3.** FRAP experiment data set 3.
DOI: https://doi.org/10.7554/eLife.41517.018

of membrane association for the two allotypes of CHC22 and for CHC17 was assessed by Fluorescence Recovery After Photobleaching (FRAP). To assess clathrin turnover, as an indicator of clathrin coat stability, cells expressing fluorescent proteins were photobleached in the perinuclear area (*Figure 7B*) and their rate of fluorescence recovery was measured. Recovery of CHC17 fluorescence was the fastest, consistent with its more soluble properties compared to CHC22 (*Dannhauser et al.,*

*2017*). CHC22-M1316 showed the slowest recovery and CHC22-V1316 was intermediate (*Figure 7C–E*), suggesting that it is more exchangeable in the CHC22 coat than the M1316 allotype.

The impact of CHC22 variation on GLUT4 retention was then assessed. Because HeLa cells express CHC22, they can form a GSC, when transfected to express GLUT4. These cells sequester GLUT4 intracellularly, and then release it to the plasma membrane in response to insulin, behaving like muscle and adipocytes, though with more modest insulin response (*Camus et al., 2018*; *Trefely et al., 2015*; *Haga et al., 2011*). To detect GLUT4 release to the cell surface, we used a construct expressing GLUT4 tagged with mCherry and a hemagglutinin (HA) epitope embedded in an exofacial loop of the transporter (HA-GLUT4-mCherry). Appearance of surface GLUT4 in response to insulin was detected by fluorescence-activated cell sorting (FACS) using an antibody to the HA epitope (*Figure 8A*). Transfection of HeLa cells with siRNA depleting CHC22 ablates this insulin-responsive pathway (*Camus et al., 2018*) (*Figure 8A*). We then assessed if siRNA inhibition of insulin-responsive GLUT4 release can be rescued by expression of CHC22-M1316-GFP or CHC22-V1316-GFP. These constructs, the same as characterized in *Figure 7A*, are siRNA-resistant, as well as being GFP-tagged. We observed that, when endogenous CHC22 was depleted, CHC22-M1316 was able to restore the insulin response but CHC22-V1316 was not, when the rescue constructs were expressed at the same levels in cells (measured by intensity of GFP fluorescence) (*Figure 8A*). CHC17 expression also did not rescue insulin-induced GLUT4 expression, as shown previously

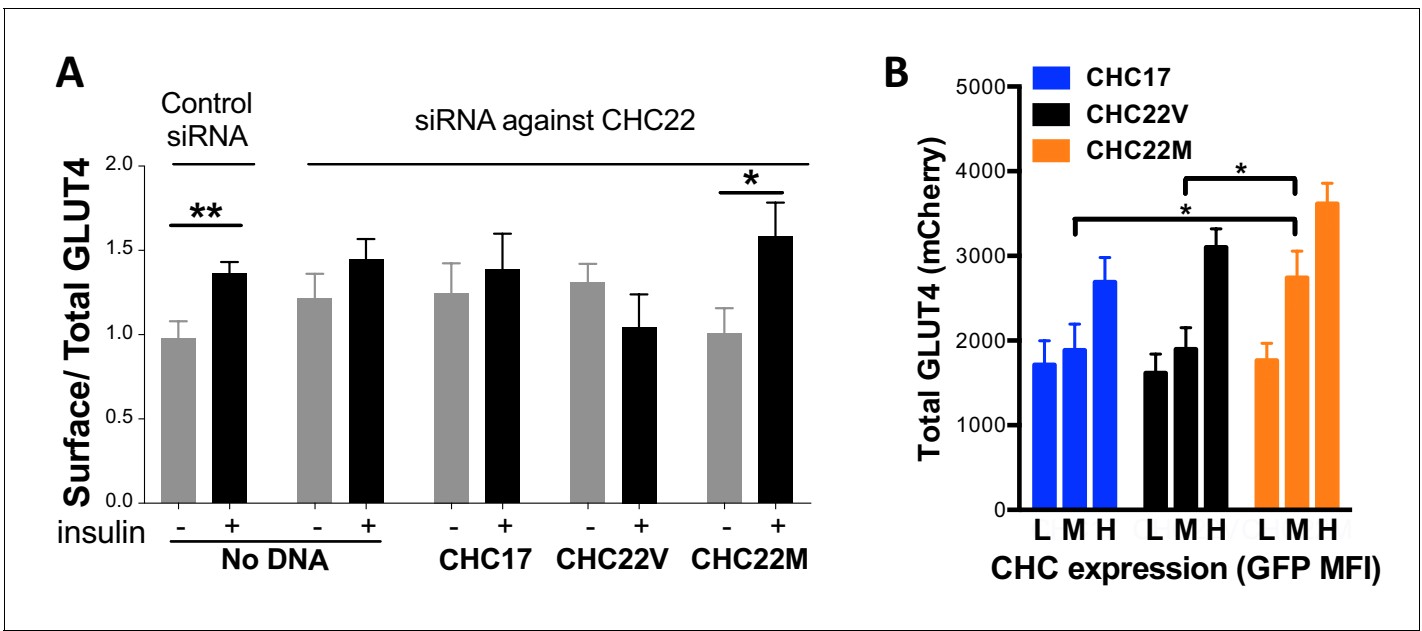

**Figure 8.** Differences in intracellular GLUT4 sequestration and stability occur in cells expressing the CHC22-M1316 or CHC22-V1316 allotypes. HeLa cells were treated with siRNA to deplete endogenous CHC22 or with control siRNA, then transfected to co-express HA-GLUT4-mCherry along with CHC17-GFP (CHC17), CHC22-M1316-GFP (CHC22M) or CHC22-V1316-GFP (CHC22V) (**A** and **B**). Total levels of expressed GLUT4 and CHC were measured by FACS (mean fluorescence intensity (MFI) for mCherry or GFP, respectively). Surface levels of GLUT4 were measured with anti-HA antibody at basal conditions (-) or after 30 min of exposure to insulin (+) and surface/total GLUT4 is reported as a measure of GLUT4 translocation to the cell surface (**A**) in cells expressing equivalent total levels of CHC-GFP. The extent of GLUT4 translocation was assessed in each experimental group before and after insulin stimulation; Student t-test, * p-value<0.05. Transfected cells treated with siRNA to deplete endogenous CHC22, but not treated with insulin, were gated into thirds expressing equivalently low (L), medium (M) and high (H) levels of CHC-GFP for each type of CHC, then total levels of HA-GLUT4-mCherry in each population were plotted (**B**). We performed a one-way analysis of variance (ANOVA) with Tukey's multiple comparison post-hoc test: * p-value<0.05.

DOI: https://doi.org/10.7554/eLife.41517.019

The following source data is available for figure 8:

**Source data 1.** GLUT4 translocation experiment.
DOI: https://doi.org/10.7554/eLife.41517.020
**Source data 2.** Total GLUT4-mCherry levels.
DOI: https://doi.org/10.7554/eLife.41517.021

(*Vassilopoulos et al., 2009*). However, CHC22-V1316 is functional for trapping GLUT4 intracellularly because CHC22-transgenic mice that express CHC22-V1316 in muscle, using the natural human promoter, show excessive GLUT4 sequestration in muscle compared to wild-type mice without CHC22, leading to higher blood glucose in the transgenic animals (*Vassilopoulos et al., 2009*). To analyze GLUT4 sequestration in another way, cells depleted for CHC22 and then transfected with mCherry-GLUT4 plus either CHC22 allotype or CHC17 were each divided into three populations expressing equivalently low, medium and high levels of the transfected CHC-GFP. Then, the total GLUT4 content of the cells was measured by mCherry fluorescence. We observed higher levels of GLUT4 in CHC22-depleted cells expressing CHC22-M1316-GFP, compared to cells expressing either CHC22-V1316-GFP or CHC17-GFP at both medium and high levels of CHC expression (*Figure 8B*). This suggests that GLUT4 is sequestered more effectively from degradative membrane traffic pathways when trafficked by CHC22-M1316 than by CHC22-V1316, indicating that the M1316 variant is more efficient at targeting GLUT4 to the GSC. As indicated by their weak insulin response compared to muscle or fat cells, HeLa cells are only just able to form a functional GSC from which GLUT4 can be released. For these cells, the less effective CHC22-V1316 is inadequate to restore GSC formation when their endogenous CHC22-M1316 is depleted. Use of this HeLa model was necessitated by the lack of natural models for the CHC22-dependent GLUT4 pathway in myoblasts and adipocytes, as well as a lack of antibodies that detect surface GLUT4. Nonetheless, these experiments demonstrate a functional difference between CHC22-M1316 and CHC22-V1316 and suggest that CHC22-V1316 is less efficient at GLUT4 sequestration.

## Discussion

We studied the phylogenetics and population genetics of CHC22 clathrin to understand the functional variation of this protein in relation to its evolutionary history. CHC22 clathrin is a key player in post-prandial blood glucose clearance in humans through its role in intracellular packaging of the GLUT4 glucose transporter in muscle and fat, the tissues in which CHC22 and GLUT4 are expressed (*Vassilopoulos et al., 2009*). The CHC22 pathway positions GLUT4 for cell surface release in response to insulin and consequent uptake of glucose into these tissues (*Bryant et al., 2002*). The *CLTCL1* gene encoding CHC22 resulted from gene duplication that we have now dated to 494–451 MYA, early in vertebrate evolution when jawed vertebrates emerged. We had previously shown that *CLTCL1* is a pseudogene in mice (*Wakeham et al., 2005*). Expanding analysis to 56 jawed vertebrate genomes (>5X coverage) we could not detect *CLTCL1* in nine of them. Six of these absences can be ascribed to two independent gene loss events in branches of the Rodentia and the Cetartidactylae. The three others may represent additional gene losses or incomplete genome annotation. All vertebrate and non-vertebrate eukaryotes considered here have retained the parent *CLTC* gene encoding CHC17 clathrin, which mediates endocytosis and other housekeeping membrane traffic pathways. The analysis described here establishes that *CLTC* is under strong purifying selection. Notable is our evidence for purifying selection on *CLTCL1* in the species in which it has been retained, supporting its functional importance in those species. Compared to *CLTC*, extensive allelic diversity was observed for *CLTCL1* in all species for which populations were analyzed, including humans, chimpanzees and bears. Variant alleles were species-specific in most cases. In all human populations, two allelic variants of *CLTCL1* are present in high frequency, differing only at one nucleotide, resulting in CHC22 protein with either methionine or valine at position 1316. The V1316 allotype appears specific to humans, but some non-human primates have a different variation at the position 1316. Analysis of ancient humans dated the appearance of the V1316 variant to 500–50 KYA and indicated that M1316, which is fixed in CHC17 clathrin, is the ancestral state. Analyses of human population genetic data provided support for the maintenance of high genetic diversity and two allotypes of CHC22. We hypothesize that selective pressure on CHC22 clathrin comes from its role in nutrient metabolism. Consistent with this hypothesis, we observed functional differences between the two CHC22 allotypes in their capacity to control GLUT4 membrane traffic, as predicted by structural modeling and differences in cellular dynamics of the two allotypes.

Retention of *CLTC* in all vertebrate species is consistent with the encoded CHC17 mediating cellular housekeeping clathrin functions shared by all eukaryotes. On the other hand, CHC22, encoded by the paralogous gene *CLTCL1*, operates in the specialized insulin-responsive GLUT4 pathway to make the pathway more efficient in those species that retained *CLTCL1*. Data presented here

(*Figure 8*) and our recent mapping of a novel intracellular location for CHC22 function (*Camus et al., 2018*) indicate that, in human cells, CHC22 clathrin promotes transport from the secretory pathway to the insulin-responsive GSC. This CHC22 pathway complements the endocytic pathway for GLUT4 targeting to the GSC, so species without CHC22 can rely primarily on endocytosis for GLUT4 trafficking to the GSC, while species with CHC22 use both pathways. Thus, we hypothesize that species with functional CHC22 clathrin are more efficient at intracellular GLUT4 sequestration, resulting in lower surface GLUT4 in the absence of insulin, and tighter regulation of GLUT4 release in response to insulin. The trade-off is that these species have an inherent increased tendency to insulin resistance as their GLUT4 is sequestered more effectively. The two main vertebrate branches that have lost CHC22 comprise the Muridae (mice and rats) who are incessant herbivores and the Cetartiodactyla (sheep, cattle, porpoise and pigs) which include the ruminants (sheep and cattle) whose muscle uptake of glucose is critical for muscle function, but is not a main pathway for glucose clearance (*Hocquette et al., 1995*). These two groups of species require greater availability of GLUT4 on their cell surfaces, so that more efficient GLUT4 sequestration by CHC22 would not be favorable to their nutritional needs. The fact that CHC22 alters the balance of membrane traffic to the GSC means that species losing *CLTCL1* could evolve compensatory pathways more compatible with their diets. Thus, transgenic mice expressing CHC22 over-sequester GLUT4 in their muscle and develop hyperglycemia with aging (*Vassilopoulos et al., 2009*). The cave fish, which appears to lack *CLTCL1,* has independently evolved mutations in the insulin receptor, creating natural insulin resistance, such that the presence of CHC22 on top of this mechanism might be detrimental (*Riddle et al., 2018*). The loss of *CLTCL1* from cave fish is consistent with the insulin responsive GLUT4 pathway being a target for natural selection driven by diet, which might also explain *CLTCL1* variation or loss for additional vertebrate species during vertebrate evolution.

The allelic variation reported here for *CLTCL1* in human and bear populations further supports the hypothesis that *CLTCL1* has undergone continued selection during vertebrate evolution in relation to diet. While purifying selection appears to be operating on *CLTCL1* in those species that retain it, *CLTCL1* is far more variable than *CLTC* in these species. In humans, we find two major and functionally distinct alleles at remarkably similar frequencies in all populations studied. Statistical analysis comparing early farmer and hunter-gatherer populations shows an apparent increase of the V1316 variant, suggesting a correlation with regular consumption of digestible carbohydrate. Notably, the SNP distinguishing these alleles is human-specific and likely arose 550–50 KYA (i.e. post-Neanderthal split, pre-Neolithic). Other dramatic increases in digestible carbohydrate utilization have been inferred for humans in this timeframe; in particular the advent of cooking (which gelatinizes crystalized starch, making it much easier to digest), salivary amylase gene copy number increase (allowing increased starch digestion capacity) and accelerated brain size increase (which would increase demands for blood glucose) (*Hardy et al., 2015*). While the co-evolution of these cultural and genetic traits was originally proposed to have occurred some 800 KYA, recent studies indicate a time frame of 450–300 KYA years for cooking (*Shahack-Gross et al., 2014*), increased oral amylase activity (*Inchley et al., 2016*) and accelerated brain size increase (*Dunbar, 2019*). The fact that the two major human *CLTCL1* alleles are functionally distinct is consistent with diversifying selection operating on *CLTCL1,* with a balancing selection possibly caused by heterozygote advantage. While population genetic signatures for balancing or overdominant selection were not entirely robust, some summary statistics were suggestive of an increased diversity that was unlikely to have occurred under neutrality. Other statistics, such as the ones based on allele frequencies, would not be expected to gain significance within the timeframe of the human-specific diversifying selection we detect. The allelic diversity of *CLTCL1* in other primate species could have the potential effect of diluting its function. Whilst chimpanzees are omnivores and gorillas herbivores, both rely for nutrition on extensive foraging for carbohydrate. Also notable is that polar bears, who have a very low carbohydrate diet compared to their brown bear relatives, have distinct CHC22 variants with unknown functionality, again consistent with *CLTCL1* undergoing selection driven by nutritional ecology.

Clathrins are self-assembling proteins and function as a latticed network in the protein coat that they form on transport vesicles. Our structural modeling predicts that the single amino acid difference between the two main human CHC22 allotypes could influence the strength of molecular interactions in the CHC22 clathrin lattice, as position 1316 occurs at a lattice assembly interface (*Figure 6*). When expressed in cells, both CHC22 variants gave the same overall intracellular distribution, but CHC22-V1316 shows faster turnover from membranes than CHC22-M1316 (*Figure 7*) and

is less effective at GLUT4 sequestration (*Figure 8B*). These properties are consistent with the methionine to valine change attenuating GLUT4 retention. This interpretation is further supported by a GLUT4 translocation assay, which indicates that the V1316 variant is less effective in forming the insulin-responsive GSC than the ancestral M1316 form of CHC22 (*Figure 8A*). Thus, mixed lattices occurring in heterozygous individuals, potentially reflect balancing selection and overdominance, might reduce GLUT4 sequestration compared to M1316 homozygotes. This would have the effect of improving glucose clearance. It can be argued that human consumption of digestible carbohydrate on a regular basis (*Hardy et al., 2015*), requiring increased glucose clearance, might be a selective force driving this genetic adaptation. This view is consistent with the increased frequency of the V1316 variant in early farmers. It is also possible that some forms of polar bear CHC22 are superactive at GLUT4 sequestration, providing a route to maintain high blood glucose, as occurs through other mutations in the cave fish (*Riddle et al., 2018*).

Regulators of fundamental membrane traffic pathways have diversified through gene duplication in many species over the timespan of eukaryotic evolution. Retention and loss can, in some cases, be correlated with special requirements resulting from species differentiation, such as the extensive elaboration of genes in the secretory pathway of Tetrahymena (*Dacks and Robinson, 2017*; *Bright et al., 2010*). The evolutionary history of *CLTCL1*, following vertebrate-specific gene duplication, suggests that differentiation of nutritional habits has shaped selection for the presence and absence of *CLTCL1* in some vertebrate species, and its diversification in humans and potentially other species. Though its highest expression is in muscle and adipose tissue, transient expression of CHC22 during human brain development has also been documented (*Nahorski et al., 2015*). This was noted in a study of a very rare null mutant of *CLTCL1* that caused loss of pain sensing in homozygotes and no symptoms for heterozygotes (*Nahorski et al., 2015*). Attenuated CHC22 function of the V1316 variant might lead to a spectrum of pain sensing in humans but this is unlikely to be a strong selective force affecting reproductive success, whereas glucose homeostasis, as suggested by our analysis, is more likely. By exerting efficient control of blood glucose levels, the presence of CHC22 clathrin was likely beneficial in providing the nutrition required to develop the large human brain, as well as affecting reproduction by influencing glucose availability during pregnancy (*Hardy et al., 2015*). However, over the last 12,500 years in association with farming, or perhaps over the last 450,000 years in association with cooking, salivary amylase activity and starch digestion (*Hardy et al., 2015*; *Shahack-Gross et al., 2014*; *Inchley et al., 2016*), readily available carbohydrate has increased our need to clear glucose from the blood, such that selection continues to act on *CLTCL1* in humans. Our cell biology studies have also demonstrated that CHC22 increases GLUT4 retention. While we would not expect the major *CLTCL1* polymorphism to directly influence the development of T2D, CHC22 accumulates on the expanded GSC that forms in cases of insulin-resistant T2D (*Vassilopoulos et al., 2009*), so its variation could potentially exacerbate insulin resistance to different degrees. The genetic diversity that we report here may reflect evolution towards reversing a human tendency to insulin resistance and have relevance to coping with increased carbohydrate in modern diets.

## Materials and methods

### Key resources table

| Reagent type (species) or resource | Designation | Source or reference | Identifiers | Additional information |
|---|---|---|---|---|
| Cell line (human) | HeLa | ATCC | Cat. #: CCL-2; RRID:CVCL_0030 | |
| Antibody | Mouse monoclonal anti-CHC17 (X22) | Frances Brodsky PMID: 2415533 | | IF (5 mg/mL) |
| Antibody | Mouse monoclonal anti-CHC17 (TD.1) | Frances Brodsky PMID: 1547490 | | WB (1.3 mg/mL) |
| Antibody | Rabbit polyclonal anti-CHC22 (SHL-KS) | Frances Brodsky PMID: 29097553 | | WB (0.4 mg/mL) |

*Continued on next page*

*Continued*

| Reagent type (species) or resource | Designation | Source or reference | Identifiers | Additional information |
|---|---|---|---|---|
| Antibody | Mouse monoclonal anti-β-actin (AC-15) | Sigma | Cat. #: A1978; RRID:AB_476692 | WB (1:2000) |
| Antibody | Purified anti-HA.11 (16B12) | Covance | Cat. #: MMS-101P; RRID:AB_10064068 | |
| Antibody | Rabbit polyclonal anti-CHC22 | Proteintech | Cat. #: 22283–1-AP; RRID:AB_11183764 | |
| Antibody | Goat anti-rabbit IgG coupled to HRP | Thermo Fisher Scientific | Cat. #: 172–1019 | WB (1:8000) |
| Antibody | Goat anti-mouse IgG coupled to HRP | Thermo Fisher Scientific | Cat. #: 170–6516 | WB (1:8000) |
| Antibody | Anti-mouse IgG1 coupled to Brilliant Violet 421 (RMG1-1) | Biolegend | Cat. #: 406616; RRID:AB_2562234 | FC (1:200) |
| Recombinant DNA reagent | HA-GLUT4-mCherry | This paper | | Generated from HA-GLUT4-GFP (gift from Dr Tim McGraw, PMID: 11058093) |
| Recombinant DNA reagent | CHC22V (pEGFP-C1-GFP-CHC22V) | Frances Brodsky PMID: 20065094 | | |
| Recombinant DNA reagent | CHC22M (pEGFP-C1-GFP-CHC22M) | This paper | | Generated by Quick change mutagenesis from CHC22V |
| Recombinant DNA reagent | CHC17 (pEGFP-C1-GFP-CHC17) | Frances Brodsky PMID: 29097553 | | |
| Sequence-based reagent | AllStars Negative Control siRNA | Qiagen | Cat. #: SI03650318 | |
| Commercial assay or kit | Quick change mutagenesis | New England Biolabs, USA | Cat. #: E0554S | |
| Commercial assay or kit | BCA | Pierce | Cat. #: 23225 | |
| Commercial assay or kit | Western Lightning Chemilumi-nescence Reagent | GE Healthcare | Cat. #: RPN2209 | |
| Chemical compound, drug | JetPrime transfection reagent | PolyPlus | Cat. #: 114–07 | |
| Chemical compound, drug | Insulin | Sigma | Cat. #: I9278 | |
| Chemical compound, drug | Bovine serum albumin (BSA) | Sigma | Cat. #: A7906 | |
| Software, algorithm | FlowJo | Treestar | | |
| Software, algorithm | ImageJ | NIH | | |
| Software, algorithm | Prism | Graphpad | | |

*Continued on next page*

*Continued*

| Reagent type (species) or resource | Designation | Source or reference | Identifiers | Additional information |
|---|---|---|---|---|
| Software, algorithm | R | R Project | | Packages: pegas, Smisc, gplots |
| Other | CellView glass bottom culture dish | Greiner Bio-one | Cat. #: 627860 | |
| Other | Nitrocellulose membrane | Biorad | Cat. #: 1620112 | |

## Phylogenetics

Vertebrate genomes as well as genomes of *Drosophila melanogaster*, *Caenorhabditis elegans*, *Ciona intestinalis* and *Ciona savignyi* were downloaded from Ensembl (*Yates et al., 2016*), all accessed on 23/04/2016 except for pig (14/12/2017), marmoset and hagfish (both 09/08/2018), excluding vertebrate species sequenced below five-fold genome coverage, that is with less than five reads per site on average. In addition, we downloaded the genomes of the elephant shark (*Venkatesh et al., 2014*), whale shark (*Read et al., 2017*), marmot (The Alpine Marmot Genome, BioProject PRJEB8272 on NCBI) and porpoise (*Yuan et al., 2018*). All potential orthologs for the human isoforms of *CLTC/CLTCL1*, *MTMR4/MTMR3*, and *CLTA/CLTB*, in the above genomes were retrieved via BLAST (*Boratyn et al., 2013*). An *e*-value threshold of 0.001 with additional constraints applied by default in InParanoid version 7 (*Ostlund et al., 2010*) were used (at least 50% of the sequences are covered by the alignment, and at least 25% of the residues are aligned). The polar bear (*Ursus maritimus*) (*Liu et al., 2014*), brown bear (*Ursus arctos*) (*Benazzo et al., 2017*) and black bear (*Ursus americanus*) CHC17 and CHC22 protein sequences were manually added. For *CLTCL1* only the elephant and horse sequences (XP_023397213.1 and XP_023502410.1 respectively) were manually added.

The sequences corresponding to the longest transcripts were aligned with MAFFT (*Katoh and Standley, 2013*) and phylogenetic trees generated with PhyML (*Guindon et al., 2010*). The last two steps were repeated after manually removing outlier sequences lying on long branches (*CLTC/CLTCL1*: ENSTNIP00000007811.1, ENSTGUP00000014952.1, XP_023397213.1, XP_023502410.1; *CLTA/CLTB*: ENSPSIP00000012669.1) and, in the case of genomes not retrieved from Ensembl (therefore lacking the gene-to-transcript mapping), sequences most likely corresponding to alternative transcripts (XP_015350877.1, XP_007899998.1, XP_007899997.1, XP_007904368.1, XP_007904367.1, XP_020375861.1, XP_020375865.1, XP_020375862.1, XP_020392037.1, XP_020375864.1, XP_020375859.1). Trees were manually reconciled based on the Ensembl species tree extended by elephant shark, whale shark, brown bear, black bear, porpoise and marmot with TreeGraph (*Stöver and Müller, 2010*). Branch lengths were estimated based on the multiple sequence alignment (MSA) with PhyML fixing the manually reconciled topology, with options '-u' and '—constraint_file'. With this approach no support values for splits are calculated. The resulting trees were used as input to generate a new phylogeny-aware MSAs with PRANK (*Löytynoja and Goldman, 2005*). Branch lengths of the reconciled topologies were then re-estimated based on the MSA generated by PRANK.

To compute evolutionary rates, the sequences and subtrees corresponding to *CLTC* and *CLTCL1* clades after duplication (i.e. excluding non-vertebrates) were extracted and sequences from species without either *CLTC* or *CLTCL1* were removed. The same procedure was performed for *MTMR4/MTMR3* and *CLTA/CLTB*. A phylogeny-aware MSA was computed with PRANK on the remaining sequences, and the amino acid alignment was converted to a codon alignment with PAL2NAL (*Suyama et al., 2006*). Finally, dN/dS ratios (i.e. the ratio of the rate of nonsynonymous substitutions to the rate of synonymous substitutions) were inferred based on the codon alignments with PAML (*Yang, 2007*) for the six proteins independently using the site model M7. Model M7 fits a Beta-distribution to the site rates by estimating the two Beta parameters shape and scale. Rates are estimated per site over the entire phylogeny, and therefore represent time averages. Phylogenetic trees of consensus amino acid sequences for bear samples only were computed using PhyML 3.1 (*Guindon et al., 2010*) with default values as implemented in Phylogeny.fr (*Dereeper et al., 2008*).

## Population genetics

Phased genotypes were obtained by querying Variant Call Format (VCF) files (*Danecek et al., 2011*) from the 1000 Genomes Project database Phase 3 (1000 *Auton et al., 2015*) for all available 2504 samples. Only high-quality variants were retained using *vcflib* (https://github.com/vcflib/vcflib) with options 'VT = SNP and QUAL > 1 and AC >1 and DP >5000'. Missing genotypes were assigned to homozygotes for the reference alleles. Finally, only sites with a recorded annotated function of being missense, nonsense, stop-loss or frame-shift for tested genes according to the UCSC Table Browser were retained (*Speir et al., 2016*) (tables snp150 and snp150CodingDbSnp). For each retained position, the reference sequence for chimpanzee from the UCSC Table Browser (*Speir et al., 2016*) (table snp150OrthoPt5Pa2Rm8) was initially used to infer the putative ancestral state. For ambiguous or multiallelic states in the chimpanzee sequence, the human reference base was used as an initial proxy for the ancestral state. The predicted functional impact of amino acid replacements was obtained by using Polyphen (*Adzhubei et al., 2010*) and SIFT (*Kumar et al., 2009*). Additional frequency information for a single mutation of interest in more than 50 human populations was retrieved from the HGDP CEPH Panel (*Cann et al., 2002*) from http://hgdp.uchicago.edu/cgi-bin/gbrowse/HGDP/. Genotype data for farmer and hunter-gatherer individuals were collected from the Simons Genome Diversity Project Dataset (*Mallick et al., 2016*). Populations were merged based on their assigned geographical region with the following classification for hunter-gatherers: Africa (Biaka, Ju|'hoan North, Khomani San, Mbuti), Central Asia and Siberia (Aluet, Chukchi, Eskimo Chaplin, Eskimo Naukan, Eskimo Sireniki, Even, Itelman, Tlingit, Tubalar, Ulchi, Atayal), East and South Asia (Atayal, Kusunda). Farmer and hunter-gatherer allele frequencies were compared following a previously described approach (*Raineri et al., 2014*). Briefly, we analytically computed the probability that the V allele is more frequent in farmers than in hunter gatherers while fully accounting for the uncertainty in the individual frequency estimates. V allele frequencies were inferred from allele counts of M and V in a Bayesian framework with a conjugate Beta uniform prior. We recorded maximum *a posteriori* estimates with 95% highest posterior density credible intervals computed with the *Smisc* R library, version 0.3.9. We collected further published ancient DNA data from Western Eurasia and classified into three genetic grouping: hunter-gatherer (HG), early farmer (EF) and steppe, using supervised ADMIXTURE (*Alexander et al., 2009*) as previously described (*Mathieson and Mathieson, 2018*). These are genetic groups and not directly based on differences in material culture or subsistence, but importantly in the case of HG and EF, these genetic classifications correspond closely to hunter-gatherer and agricultural subsistence strategies (*Haak et al., 2015*; *Skoglund et al., 2014*; *Skoglund et al., 2012*). We then restricted analysis to samples dated between 10,000 and 5,000 years before present that were classified as either HG or EF, leading to a dataset of 119 HG and 316 EF of which 85 and 188 respectively had coverage at rs1061325. Frequencies for South-East Asians and ancient Eurasians were down-sampled to ensure numerical stability. The HeLa genomic data were accessed through the NIH database of Genotypes and Phenotypes (dbGaP at http://www.ncbi.nlm.nih.gov/sites/entrez?db=gap) through dbGaP accession number phs000640.

High-coverage VCF files for 79 individuals from six species of great apes were retrieved (*Prado-Martinez et al., 2013*). Data was filtered using *vcflib* on the combined data set with the options 'QUAL > 32 and DP >50 and DP <7000 and FS <27 and MQ > 25 and AC >1', similarly to the original manuscript describing this data set (*Prado-Martinez et al., 2013*). To retrieve nonsynonymous changes, only variants where the translated proteins for each allele differ were retained. We finally phased the data and assigned individual haplotypes using *shapeit* v2.r837 with the options '-burn 50 -prune 20 -main 100 -window 0.5 -effective-size 20000'. Additional 110 genomes of chimpanzees and bonobos were analyzed (*Teixeira et al., 2015*; *de Manuel et al., 2016*). Data filtering, functional annotation and haplotype phasing were performed as described above.

Full genome VCF files for two high-coverage archaic humans, namely one Altai Neanderthal (*Prüfer et al., 2014*) and one Denisova were retrieved (*Meyer et al., 2012*). Low-quality sites were filtered out using *vcflib* with the options 'QUAL > 1 and DP >10'. A pseudo-reference sequence for each archaic human was constructed by replacing the heterozygous sites with the previously inferred human ancestral state. Sequencing data information for additional ancient human samples were obtained from previously published high-quality whole genome sequences (*Skoglund et al., 2014*; *Broushaki et al., 2016*; *Hofmanová et al., 2016*; *Lazaridis et al., 2014*; *Olalde et al., 2014*;

*Raghavan et al., 2014*; *Seguin-Orlando et al., 2014*; *Fu et al., 2014*). Genotype likelihoods were calculated using the standard GATK model (*McKenna et al., 2010*). Median-joining network plots were generated in R using *pegas* package (*Paradis, 2010*).

Several summary statistics were calculated on the inferred alleles to describe their levels of nucleotide diversity. Specifically, for each population separately, Watterson's estimator of population mutation parameter (TW) (*Watterson, 1975*), Nei's genetic diversity index (PI) (*Nei, 1973*), Tajima's D (TD) (*Tajima, 1989*), Fu and Li's D* (FLDs) and F* (FLFs) (*Fu and Li, 1993*), the sum of squared allele frequencies including the most common allele (H1) and excluding it (H2) and their normalized ratio (H2H1) (*Garud and Rosenberg, 2015*; *Garud et al., 2015*) were calculated. We also computed genetic differentiation ($F_{ST}$) (*Reynolds et al., 1983*) between pairs of canonical reference populations, namely Yoruban (YRI), Europeans (CEU), and Han Chinese (CHB).

To assess whether the observed summary statistics are expected under neutral evolution, genes with a coding length approximately equal (±5%) to the one observed for the tested gene, *CLTCL1*, were selected. For this analysis, the longest isoform for each gene, and its annotation was considered according to refGene table from the UCSC Genome Browser. We discarded genes on chromosome six and on sex chromosomes, as well as *CLTA*, *CLTB* and *CLTC*. This set was further reduced to the first 500 genes with the closest genomic length to *CLTCL1*. As summary statistics can be calculated only in case of genetic variability, genes showing no non-synonymous SNPs within each population were discarded. For each summary statistic, the empirical percentile rank for the value observed in *CLTCL1* compared to the whole distribution of control genes was calculated. Low or high values are suggestive of *CLTCL1* being an outlier in the empirical distribution. For plotting purposes, summary statistics and populations were clustered according to a dendrogram inferred from their respective distances based on the calculated matrix of empirical percentile ranks. That is, populations clustering together exhibit similar patterns of percentile ranks, and thus of summary statistics. The underlying dendrograms are not reported. The heatmap plot was generated using the function heatmap.2 in R with the package *gplots*. Cells with an empirical percentile rank lower than 0.10 or greater than 0.90 were filled with the exact rank value. We also obtained a null distribution of summary statistics by performing coalescent simulations using *msms* (*Ewing and Hermisson, 2010*) under a previously derived demographic model for human populations (*Gutenkunst et al., 2009*).

## Structure prediction by modeling

MODELLER v9.13 (*Benjamin and Sali, 2014*) was used to model the structure of the proximal leg segment of CHC22, using the crystal structure of bovine CHC17 (PDB 1B89) (*Ybe et al., 1999*) as a template. The model of the M1316V mutant was derived in a similar way using a mutated sequence. Structure visualization and analysis of residue interactions at the mutation site M1316 were performed using UCSF Chimera (*Pettersen et al., 2004*). The wild type and mutant homology models were positioned in the cryo-electron microscopy map of the bovine clathrin lattice (EMD: 5119) (*Fotin et al., 2004*) by structural superposition on the atomic model originally fitted in the map (PDB 1XI4).

## Functional experiments

### Antibodies, plasmids and reagents

Mouse monoclonal anti-CHC17 antibodies X22 (*Brodsky, 1985*), TD.1 (*Näthke et al., 1992*) and affinity-purified rabbit polyclonal antibody specific for CHC22 and not CHC17 (*Vassilopoulos et al., 2009*) were produced in the Brodsky laboratory. Commercial sources of antibodies were as follows: mouse monoclonal anti-β-actin (clone AC-15, Sigma), mouse monoclonal anti-HA (clone 16B12, Covance), rabbit polyclonal anti-CHC22 (Proteintech). Secondary antibodies coupled to HRP were from ThermoFisher, the secondary antibody coupled to Brilliant Violet 421 was from BioLegend. The HA-GLUT4-mCherry was generated by replacing the GFP from the HA-GLUT4-GFP construct (gift from Dr Tim McGraw; *Lampson et al., 2000*) with mCherry using KpnI and EcoRI. The generation of the CHC22 variant expressing a valine at position 1316 (CHC22V) was previously described (*Esk et al., 2010*). The CHC22 variant expressing a methionine at position 1316 (CHC22M) was generated from CHC22V by quick-change mutagenesis (New England Biotechnologies, USA) following manufacturer's instructions.

## Small RNA interference

Targeting siRNA was produced to interact with DNA sequences AAGCAATGAGCTGTTTGAAGA for CHC17 (*Esk et al., 2010*) (Qiagen), TCGGGCAAATGTGCCAAGCAA and AACTGGGAGGATCTAG TTAAA for CHC22 (1:1 mixture of siRNAs were used) (*Vassilopoulos et al., 2009*) (Dharmacon). Non-targeting control siRNA was the Allstars Negative Control siRNA (Qiagen).

## Cell culture

HeLa cells were grown in Dulbecco's Modified Eagle Medium high glucose (Gibco) supplemented with 10% FBS (Gibco), 50 U/mL penicillin, 50 µg/mL streptomycin (Gibco), 10 mM Hepes (Gibco) and maintained at 37°C in a 5% $CO_2$ atmosphere. HeLa cells were free of mycoplasma infection.

## siRNA and DNA transfection

Cells were transfected for 72 hr with 20 nM of siRNA. Silencing was assessed by immunoblotting. Transient DNA transfections for rescue experiments were performed during the third day of silencing. For FACS experiments, cells (per well of 6-well plate, 70% confluent) were transiently transfected with 1 µg DNA for CHC22M-GFP and CHC22V-GFP, 1.5 µg DNA for CHC17-GFP and HA-GLUT4-mCherry. For FRAP experiments, cells (per glass bottom dish, 60% confluent) were transfected with 0.75 µg DNA for CHC22-GFP (M or V) or 1.5 µg DNA for CHC17-GFP. FACS and FRAP experiments were carried out 24 hr later. All transfections were performed using JetPrime transfection reagent (PolyPlus) following manufacturer's instructions.

## GLUT4 translocation assay using flow cytometry

HeLa cells were grown in 6-well plates and transiently transfected with either HA-GLUT4-mCherry alone or in combination with GFP-tagged CHC22 (M or V) or CHC17-GFP the day before the experiment. The next day, cells were serum-starved (2 hr) before insulin stimulation (170 nM or vehicle (water) for 15 min, 37°C). Cells were then placed on ice and rapidly washed (2X, PBS, 4°C) and fixed (PFA 2%, 15 min). After fixation, cells were washed (2X, PBS, RT) then blocked for 1 hr (PBS 2% BSA, RT) before incubation with monoclonal anti-HA antibody (45 min, RT) to detect surface GLUT4. After incubation, cells were washed (3X, PBS, RT) and incubated with anti-mouse secondary Ig coupled to Brilliant Violet 421 (45 min, RT). Cells were then washed (5X, PBS, RT), gently lifted using a cell scraper (Corning), pelleted (800xg, 8 min) and re-suspended (PBS, 2% BSA, 4°C). Data was acquired with Diva acquisition software by LSRII flow cytometer (Becton Dickinson). Typically, 10,000 to 30,000 events were recorded and Mean Fluorescence Intensity (MFI) values for surface GLUT4 (Brilliant Violet 421) and total GLUT4 (mCherry) were recorded using 450/50 and 530/30 filters, respectively. The ratio of surface to total MFI was calculated to quantify the extent of GLUT4 translocation. MFI values for total GLUT4 (mCherry) were plotted for GLUT4 stability assays. Histograms and post-acquisition analysis were performed using FlowJo software (Treestar). Total GLUT4 and surface GLUT4 values are reported separately for cells expressing CHC22 variants with equalized GFP signals at the top third (high), middle third (medium) and bottom third (low) levels of expression.

## Fluorescence Recovery After Photobleaching

The imaging of transiently transfected HeLa cells grown on Cellview glass bottom culture dishes (Greiner, Germany) was performed at 37°C in a 5% $CO_2$ atmosphere, using low 488 nm laser power to minimize photobleaching using a 63x (1.4 NA) lens on a Leica SP8 confocal microscope. A 2.0 µm² circular region of interest was positioned in the perinuclear region of the transfected cells, a region where both CHC22 and CHC17 naturally occupy. A 100% laser power (488 nm) coupled to 11 iterations was performed to achieve GFP photobleaching. Recovery of fluorescence was recorded from 4 to 10 independent cells per dish. The experiment was repeated at least three times.

## Immunoblotting

HeLa cells protein extracts were quantified by BCA (Pierce), separated by SDS-PAGE (10% acrylamide), transferred to nitrocellulose membrane (0.2 µm, Biorad), labeled with primary antibodies (1–5 µg/mL), washed and labeled with species-specific horseradish peroxidase-conjugated secondary antibodies (ThermoFisher). Peroxidase activity was detected using Western Lightning Chemiluminescence Reagent (GE Healthcare). The molecular migration position of transferred proteins was

compared to the PageRuler Prestain Protein Ladder 10 to 170 kDa (Thermo Fisher Scientific). Signals were detected using the Chemidoc XRS + imaging system (Biorad) and quantifications were performed using Image J software (NIH).

## Statistical analyses
Graphs and statistical analyses were performed using Prism software (Graphpad). Detailed statistical information including statistical test used, number of independent experiments, *p*-values, definition of error bars is listed in individual figure legends. All experiments were performed at least three times.

## Acknowledgements
This work was supported by grants from the National Institutes of Health (USA) DK095663 (FMB and MF) and AI090905 (PP) and Medical Research Council grant MR/S008144/1 and Wellcome Trust Investigator Award 107858/Z/15/Z to FMB. We are grateful to Libby Guethlein for helpful discussions on genomic data retrieval and processing. Genomic sequence data and cell culture studies reported here utilize the HeLa cell line. Henrietta Lacks, and the HeLa cell line that was established from her tumor cells, have made significant contributions to scientific progress and advances in human health.

## Additional information

### Funding

| Funder | Grant reference number | Author |
| --- | --- | --- |
| National Institutes of Health | DK095663 | Matteo Fumagalli<br>Frances M Brodsky |
| National Institutes of Health | AI090905 | Peter Parham |
| Medical Research Council | MR/S008144/1 | Frances M Brodsky |
| Wellcome Trust | 107858/Z/15/Z | Frances M Brodsky |

The funders had no role in study design, data collection and interpretation, or the decision to submit the work for publication.

### Author contributions
Matteo Fumagalli, Stephane M Camus, Yoan Diekmann, Formal analysis, Investigation, Visualization, Methodology, Writing—review and editing; Alice Burke, Marine D Camus, Investigation, Writing—review and editing; Paul J Norman, Resources, Formal analysis, Writing—review and editing; Agnel Joseph, Investigation, Visualization, Writing—review and editing; Laurent Abi-Rached, Resources, Writing—review and editing; Andrea Benazzo, Rita Rasteiro, Resources, Data curation, Writing—review and editing; Iain Mathieson, Formal analysis, Investigation, Writing—review and editing; Maya Topf, Peter Parham, Supervision, Writing—review and editing; Mark G Thomas, Conceptualization, Supervision, Writing—review and editing; Frances M Brodsky, Conceptualization, Supervision, Funding acquisition, Writing—original draft, Project administration

### Author ORCIDs
Matteo Fumagalli (iD) https://orcid.org/0000-0002-4084-2953
Rita Rasteiro (iD) http://orcid.org/0000-0002-4217-3060
Frances M Brodsky (iD) https://orcid.org/0000-0002-1334-9258

### Decision letter and Author response
Decision letter https://doi.org/10.7554/eLife.41517.027
Author response https://doi.org/10.7554/eLife.41517.028

# Additional files

## Supplementary files

• Supplementary file 1. Human population genetics data. (**a**) Human populations from the 1000 Genomes Project analyzed with their abbreviation. (**b**) Human alleles for the coding region of *CLTCL1* extracted from the 1000 Genomes project data set. For each unique allele (hap_ID on the first column), the count of occurrences in each meta-population (EUR, EAS, AMR, SAS, AFR defined as in the legend to *Figure 3*), archaic humans (Altai Neanderthal and Denisovan) and modern chimpanzees is reported. Columns numbered 1–46 indicate the nucleotide sequence at all retrieved SNPs present in each allele. Hap-1 is the most frequent allele encoding M1316 in CHC22 and Hap-2 is the most frequent allele encoding V1316 in CHC22. (**c**) Functional annotation for each coding polymorphism of *CLTCL1* reported in *Supplementary file 1b*. Columns represent chromosome, genomic position, SNP ID, reference allele, alternate allele and functional impact. (**d**) Archaic and ancient human M1316V genotypes retrieved. Genotype likelihoods and sources for each sample are reported.

DOI: https://doi.org/10.7554/eLife.41517.022

• Supplementary file 2. Summary statistics and tests for neutrality. (**a**) Summary statistics of genetic diversity for *CLTCL1* calculated for all analyzed human populations. Abbreviations for summary statistics (on the columns) are reported in Materials and methods while abbreviations for populations (POP, on the rows) are listed in *Supplementary file 1a*. (**b**) List of 500 control genes used to assess deviation from neutrality in the summary statistics of *CLTCL1*. (**c**) Expected and observed heterozygosity for M1316 and V1316 for all analyzed human populations. For each population (Pop., on rows, with abbreviations described in *Supplementary file 1a*), the frequency of observed homozygous (Obs.Homo1 and Obs.Homo2) and heterozygous genotypes (Obs.Hetero), the corresponding expected values under Hardy-Weinberg equilibrium (Exp.Homo1, Exp.Homo2, Exp.Hetero), the ratio (RatioHetero) between observed and expected heterozygosity, and *p*-values for deviation from Hardy-Weinberg equilibrium (chi-squared test) are reported.

DOI: https://doi.org/10.7554/eLife.41517.023

• Supplementary file 3. Inferred allotypes for CHC22. (**a**) Inferred CHC22 allotypes for great apes and human genome reference. For each unique allele (hap_ID on the first column), the frequency in each species or subspecies and the amino acid sequence at each numbered position in the encoded CHC22 allotype is reported. (**b**) Inferred CHC22 allotypes for chimpanzees and bonobos. For each unique allele (hap_ID on the first column), the frequency in each species or subspecies, and the amino acid sequence at each numbered position in the encoded CHC22 allotype is reported. (**c**) Differences in the amino acid sequences for CHC22 allotypes in bears. The first row depicts the species, while the second row is the originating country for each sample. All remaining rows are the numbered positions of the polymorphic sites in CHC22 encoded in brown bears and polar bears and indicate the amino acid is present in each sample's sequence. Amino acids for the reference sequences of black bears, pandas and humans are also reported at these polymorphic positions.

DOI: https://doi.org/10.7554/eLife.41517.024

• Transparent reporting form

DOI: https://doi.org/10.7554/eLife.41517.025

## Data availability

All data generated or analysed during this study are included in the manuscript and supporting files.

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
