## [Decision Letter]

Thank you for submitting your article "The evolution of CHC22 clathrin influences its role in human glucose metabolism" for consideration by *eLife*. Your article has been reviewed by two peer reviewers, and the evaluation has been overseen by a Reviewing Editor and Patricia Wittkopp as the Senior Editor. The following individual involved in review of your submission has agreed to reveal his identity: Etienne Patin (Reviewer #2).

The reviewers have discussed the reviews with one another and the Reviewing Editor has drafted this decision to help you prepare a revised submission.

Summary:

This manuscript entitled "The evolution of CHC22 clathrin influences its role in human glucose metabolism" studies the evolution in vertebrates of the *CLTCL1* gene, which encodes the clathrin CHC22 protein. Clathrin proteins assemble to form the lattice-like coat of vesicles that enable intracellular trafficking. CHC22-coated vesicles are known to regulate the sequestration of blood glucose via GLUT4 in muscle and fat, suggesting that *CLTCL1* is under changing selection pressures related to species nutritional intake. The study reports (i) a phylogenetic analysis of the *CLTCL1* gene and its paralog *CLTC*, revealing the presence or absence of the *CLTCL1* gene in published genomes of 62 species, (ii) an estimation of evolutionary rates of *CLTCL1* and its paralog *CLTC* in vertebrates, (iii) neutrality tests based on the intraspecies genetic diversity of the human species, and (iv) the functional characterization of the M1316V *CLTCL1* variant, a non-synonymous mutation that separates the two main haplotypes observed at *CLTCL1* in humans, and is differentiated between human populations with distinct modes of subsistence. The authors conclude that *CLTC* and *CLTCL1* have evolved under purifying selection in vertebrates, but functional genetic diversity in *CLTCL1* is larger than that of *CLTC* in humans and other great apes, suggesting diversifying or balancing selection. Although the work is of interest, and the analyses and interpretation generally straightforward, the authors provide only limited evidence in support of their main thesis that *CLTCL1* evolves under balancing selection in humans. This claim needs to bolstered in the revision, or alternative hypotheses considered and discussed.

Essential revisions:

1) Balancing selection conclusion. The key pieces of populations genetics evidence offered are: (i) the *CLTCL1* gene presents a significant excess of singletons and significantly negative Tajima's D, which is indicative of positive or weak negative selection, (ii) the M1316V variant that separates the two most frequent haplotypes at the locus is differentiated across populations, while balancing selection is predicted to decrease FST (if balanced alleles have reached their equilibrium frequency in all populations), (iii) heterozygosity is both increased and decreased across human populations compared to HWE, and HWE is not tested. The authors propose no clear hypotheses that would explain overdominance of the two CHC22 forms. We suggest that the authors avoid claiming that the gene evolves under balancing selection, or provide stronger evidence for balancing selection. An intriguing observation made by the authors is the presence of seemingly shared *CLTCL1* polymorphisms between great ape species. The authors could confirm and explore this result further, as shared polymorphisms between species is a hallmark of balancing selection.

2) Alternative hypotheses. High H2H1 statistics and correlations between allele frequencies and lifestyle are more compatible with a soft sweep scenario (including multiallelic positive selection/diversifying selection and positive selection on standing variation). However, evidence for positive selection could also be questioned, and suggests instead that the gene simply evolves under weak negative selection. Namely, (i) H2 values are not statistically significant, when considering a P-value significance threshold of 5%; (ii) the highest value that H2H1 can take depends on H12 (Garud and Rosenberg, 2015), suggesting that the ratio for *CLTCL1* should be compared to random genes with similar H12 values, or should be normalized; and (iii) differences in M1316V allele frequency between hunter-gatherer and farmer populations should be compared to an empirical distribution based on random SNPs with similar global frequency, to account for population differentiation.

3) Bear species data. The analyses performed on bear species are very limited and not really informative. More importantly, no results are actually shown. Could the authors show, in a unique figure, networks for the two paralogous genes in both humans, chimps and bears? This will illustrate the systematically greater genetic diversity of *CLTCL1* relative to *CLTC*. Also, could the authors add the ancestral state in each network? This could help interpretation (under long-term balancing selection, the ancestral state should be found between the major haplotypes, contrarily to multiallelic positive selection or weak negative selection). Also could the authors describe the mutations that separate the most ancestral human haplotype from the second most frequent haplotype?

---

## [Author Response]

Essential revisions:1) Balancing selection conclusion. The key pieces of populations genetics evidence offered are: (i) the CLTCL1 gene presents a significant excess of singletons and significantly negative Tajima's D, which is indicative of positive or weak negative selection, (ii) the M1316V variant that separates the two most frequent haplotypes at the locus is differentiated across populations, while balancing selection is predicted to decrease FST (if balanced alleles have reached their equilibrium frequency in all populations), (iii) heterozygosity is both increased and decreased across human populations compared to HWE, and HWE is not tested. The authors propose no clear hypotheses that would explain overdominance of the two CHC22 forms. We suggest that the authors avoid claiming that the gene evolves under balancing selection, or provide stronger evidence for balancing selection. An intriguing observation made by the authors is the presence of seemingly shared CLTCL1 polymorphisms between great ape species. The authors could confirm and explore this result further, as shared polymorphisms between species is a hallmark of balancing selection.

We agree with the reviewers that we did not find strong evidence for balancing selection, though some statistics are suggestive. We therefore toned down such claims and present balancing selection as a possible scenario to explain our findings. Likewise, we lighten the section on overdominance and move the corresponding figure into supplementary material. As requested, we now report statistical tests for deviation from HWE and found one European population to have a nominal significant p-value towards an excess of heterozygotes. However, we do not expect large and significant deviations from HWE unless selection is very strong. We now performed neutral simulations to test against neutrality for genetic diversity (as described below) to provide further indications of non-neutrality.

We have added a functional explanation for overdominance of the two forms in humans (mixed clathrin lattices would be formed with novel properties) and we have discussed how this might be favourable for handling increased glucose in the diet from the advent of cooking, in addition to farming.

The polymorphism observed in humans is not shared by great apes, though a different polymorphism is observed at the same coding position. This suggests that variation at this position may effectively change protein function, but that the frequent polymorphism observed only in humans may result from selection due to a human-specific behaviour, e.g. cooking or farming. We have clarified this where relevant.

2) Alternative hypotheses. High H2H1 statistics and correlations between allele frequencies and lifestyle are more compatible with a soft sweep scenario (including multiallelic positive selection/diversifying selection and positive selection on standing variation). However, evidence for positive selection could also be questioned, and suggests instead that the gene simply evolves under weak negative selection. Namely, (i) H2 values are not statistically significant, when considering a P-value significance threshold of 5%; (ii) the highest value that H2H1 can take depends on H12 (Garud and Rosenberg, 2015), suggesting that the ratio for CLTCL1 should be compared to random genes with similar H12 values, or should be normalized; and (iii) differences in M1316V allele frequency between hunter-gatherer and farmer populations should be compared to an empirical distribution based on random SNPs with similar global frequency, to account for population differentiation.

Following the reviewers’ suggestion, we have now calculated the normalized H2H1 statistics (as described in Garud and Rosenberg, 2015). These are now included in the new Figure 4. Please also note that we replaced the number of polymorphisms and singletons with two more commonly used metrics of diversity: Watterson’s theta estimator (TW) and Nei’s diversity (PI). Moreover, to be more conservative, we now restrict the control distribution to the 500 genes with the most similar genomic length to *CLTCL1* (in addition to the previous requirement of having a similar coding length).

Summary statistics can be highly influenced by the number of polymorphisms used. Furthermore, we noticed that *CLTCL1* lies in a highly recombining region (according to USCS Genome Browser tables, deCODE data). Therefore, we performed an additional analysis and simulated 1,000 genomic loci of 100kbp under neutral evolution to derive a second null distribution for nucleotide diversity for Africans, Europeans and East Asians. The genetic diversity, measured by Watterson’s theta estimator (TW) and Nei’s diversity (PI), for a 100kbp genomic region (therefore including silent SNPs too) surrounding M1316V is greater than expected by drift only, suggesting an excess of genetic diversity. This is now reported in the main text. These results are robust to the length of the genomic region considered (e.g. 50k, not reported in the main text). These findings suggest that the diversity we observed for *CLTCL1* is unlikely to be observed by genetic drift only, although Tajima’s D does not appear to be statistically significant (although East Asians show higher Tajima’s D than expected by neutrality with a marginally non-significant p-value).

As suggested by the reviewers, we compared the difference in allele frequency between ancient hunter-gatherers (HG) and early farmers in a set of control SNPs with a similar global minor allele frequency than M1316V. We performed this analysis only on the set of ancient HG and early farmers, as it was the only difference which appeared to be statistically significant based on the observed data. Specifically, we sampled non-synonymous polymorphisms with similar global minor allele frequency (+/- 5%) and with a minimum global sequencing depth of 100X. We randomly retrieved 2,500 control SNPs and tested how likely was to observe a difference in allele frequency between ancient HG and early farmers equal or greater than the value observed for M1316V (+0.2658). When considering the direction of the change in derived allele frequency, we found statistical significance (p=0.036), suggesting that population structure is unlikely to explain such increase in frequency for early farmers. On the other hand, we did not find statistical support (p=0.080) when we tested for the magnitude of the change (i.e. the module of the difference). These analyses have been added to the main text. Finally, these results are robust to the choice of the control distribution. In fact, if we restrict the control SNPs to be much closer to M1316V for the global minor allele frequency (+/- 2% with 1,250 random SNPs), we obtained p-values of 0.037 and 0.082 (not reported) when testing the sign and the module, respectively.

3) Bear species data. The analyses performed on bear species are very limited and not really informative. More importantly, no results are actually shown. Could the authors show, in a unique figure, networks for the two paralogous genes in both humans, chimps and bears? This will illustrate the systematically greater genetic diversity of CLTCL1 relative to CLTC.

We are not able to provide haplotype networks for the bear species. For this data set we reconstructed a consensus amino-acid sequence for each individual and did not attempt to infer phased haplotypes due to low-coverage sequencing data, lack of reference data and limited sample size. Under these conditions, haplotypes would be phased with high uncertainty. Nevertheless, we include this data to address the reviewers’ point that no data was provided. We now present Supplementary file 3C showing all private amino-acid substitutions in polar bears or brown bears for CHC22. We also generated a phylogenetic tree for *CLTC* and *CLTCL1* using this data set and provide it as Figure 3—figure supplement 1. The tree for *CLTCL1* exhibits a greater differentiation between polar and brown bears than *CLTC*.

Following the reviewers’ suggestion, we now provide a unique figure with all 4 networks (*CLTC/CLTCL1* for humans and primates).

Also, could the authors add the ancestral state in each network? This could help interpretation (under long-term balancing selection, the ancestral state should be found between the major haplotypes, contrarily to multiallelic positive selection or weak negative selection).

Putative human ancestral sequences (e.g. from archaic and chimp sequences) were inferred only for the non-synonymous sites which are polymorphic in humans. Author response image 1 shows a network for the most common alleles (circle I has M1316 while circle II has 1316V) in humans including the chimp reference sequence (circle XIV) and archaic humans (Altai Neandertal and Denisova, embedded in circles IV and VI, respectively). Therefore, the putative human ancestral state clusters with the M1316 group, as stated in the text. In the network, please note that sizing of major alleles was down-sized (as described in the text) for plotting purposes.

Furthermore, in Author response image 2 we show a network where we also included all fixed substitutions between the 8 most common alleles in humans (light blue) and the 3 most common ones in the primate data set (red, as proxy for the ancestral state). Here, the number of substitutions between each pair of alleles is shown. Again, results indicate that the putative ancestral form is closer to M1316 allele. Please note that due to the high polymorphic levels and recombination rates, networks on the full genomic region are not informative in this case. Please also note that sizes of major alleles were reduced for plotting purposes.

**Author response image 2. respfig2:** 

These plots (not included in the revised text) illustrate what we describe there for the reviewers and confirm that the putative ancestral state clusters within the M1316 allele. We do not observe the putative ancestral allele to be in the middle of the two main human alleles, possibly because there are only non-synonymous changes between clusters and/or due to the high genetic diversity in non-human primates.

Also could the authors describe the mutations that separate the most ancestral human haplotype from the second most frequent haplotype?

There is only one mutation (M1316V) differing from the two most common alleles in the human data set. This is stated in the main text.